# Reducing errors on estimates of the carbon uptake period based on time series of atmospheric $CO_2$

Theertha Kariyathan[1,2], Ana Bastos[1], Julia Marshall[3], Wouter Peters[2], Pieter Tans[4], and
Markus Reichstein[1]

[1]Max Planck Institute for Biogeochemistry, Jena, Germany
[2]Wageningen University and Research, Netherlands
[3]Deutsches Zentrum für Luft- und Raumfahrt (DLR), Institut für Physik der Atmosphäre, Oberpfaffenhofen, Germany
[4]Institute of Arctic and Alpine Research, U. of Colorado, Boulder, CO 80309, USA

**Correspondence:** Theertha Kariyathan (tkariya@bgc-jena.mpg.de)

**Abstract.** High-quality, long time series measurements of atmospheric greenhouse gases show interannual variability in the measured seasonal cycles. These changes can be analyzed to better understand the carbon cycle and the impact of climate drivers. However, nearly all discrete measurement records contain gaps and have noise due to the influence of local fluxes or synoptic variability. To facilitate analysis, filtering and curve-fitting techniques are often applied to these time series. Previous studies have recognized that there is an inherent uncertainty associated with this curve-fitting and the choice of a given mathematical method might introduce biases. Since uncertainties are seldom propagated to the metrics under study, this can lead to misinterpretation of the signal. In this study, we use an ensemble-based approach to quantify the uncertainty of the derived seasonal cycle metrics. We apply it to $CO_2$ dry air mole fraction time series from flask measurements in the Northern Hemisphere. We use this ensemble-based approach to analyze the carbon uptake period (CUP: the time of the year when the $CO_2$ uptake is greater than the $CO_2$ release): its onset, termination and duration. Previous studies have diagnosed CUP based on the dates on which the detrended, zero-centered seasonal cycle curve switches from positive to negative (the downward zero-crossing date DZCD) and vice versa (upward zero-crossing date UZCD). However, the UZCD is sensitive to the skewness of the $CO_2$ seasonal cycle during the net carbon release period. Hence, we develop on an alternative method proposed by Barlow et al. (2015) to estimate the onset and termination of the CUP based on a threshold defined in terms of the first-derivative of the $CO_2$ seasonal cycle. Using the ensemble approach we arrive at a tighter constraint to the threshold by considering the annual uncertainty, we call this ensemble of first derivative (EFD) method. Further, using the EFD approach and an additional curve fitting algorithm, we show that (a) the uncertainty of the studied metrics is smaller using the EFD method than when approximated using the timing of the zero-crossing dates (ZCD), and (b) the onset and termination dates derived with the EFD-method provide more robust results, irrespective of the curve-fitting method applied to the data. The code is made freely available under a Creative Commons-BY license, along with the documentation in this paper (https://doi.org/10.17617/3.ZKX9JS).

# 1 Introduction

Ongoing in-situ measurements of the atmospheric $CO_2$ mixing ratio have revealed an increase in $CO_2$ mole fraction in the atmosphere. The increase in atmospheric $CO_2$ due to release of carbon from fossil fuel burning and land-use change is buffered by net $CO_2$ uptake by the ocean and land biosphere (Keeling, 1960). Since then, many studies have used high precision mea-
surements of greenhouse gases at MLO and other sites across the globe to better understand the role of $CO_2$ in global climate (e.g. Langenfelds et al., 2002; Keeling et al., 2017; Barlow et al., 2016). The analysis of such atmospheric time series helps to identify and isolate the long-term trends, inter-annual variability and seasonality of climatically important greenhouse gases (Thoning et al., 1989). However, these measurement records contain gaps and are influenced by local fluxes or synoptic scale variability, which induce noise on the underlying climate signals. Hence the use of filtering and curve-fitting techniques to
obtain smooth and continuous data has been an inevitable part of such studies (Trivett et al., 1989). The choice of mathematical method for data processing can, however introduce biases that can result in misinterpretation of the signal (Nakazawa et al., 1997; Tans et al., 1989; Pickers and Manning, 2015; Barlow et al., 2015).

Curve-fitting methods are often used to pre-process atmospheric time series for analysis. Three examples are found in the
commonly-used software packages, HPspline (Bacastow et al., 1985), CCGCRV (Thoning et al., 1989) and STL (Cleveland et al., 1990). Each of these methods produce a gap-filled time series that contains the important features of the atmospheric record, however the resultant fitted curves vary significantly from each other owing to differences in their response to gaps and outliers in the original data. Pickers and Manning (2015) addressed the sensitivity of scientific conclusions to the curve-fitting method used, by repeating a scientific study (Piao et al., 2008) using two additional curve-fitting method. Both studies looked
at changes in the $CO_2$ seasonal cycle zero-crossing date (ZCD) for ten mid-to-high-latitude, Northern Hemisphere stations. The re-analysis by Pickers and Manning (2015) found that the major conclusion of Piao et al. (2008) was robust, but that inferences at individual stations depended on the curve fitting method. This was corroborated by Barlow et al. (2015) who used a wavelet-based curve fitting method to illustrate the sensitivity of various key aspects of the seasonal cycle of $CO_2$ time series to the curve fitting approach. Thus, the impact of bias introduced by data processing methods can vary based on the data set
used and the type of analyses performed. Each method has its strengths and weaknesses; hence Pickers and Manning (2015) argued that data must be analyzed with multiple approaches to ensure that results are robust and free from bias. Despite this recommendation, studies that focus on metrics of time series such as the ZCD or seasonal cycle amplitude usually use a single curve-fitting method for analysis (e.g. Park et al., 2019; Piao et al., 2018), which can lead to differences in the conclusions that are drawn. An example is the disagreement in the direction of the trend of the $CO_2$ seasonal cycle amplitude (SCA) at Alert,
Canada between Chan and Wong (1990) and Keeling et al. (1996), as shown by Pickers and Manning (2015).

Metrics derived from $CO_2$ time series such as the seasonal cycle peaks can be highly sensitive to data gaps and noise. This is especially true for metrics associated with the growing season onset at higher latitude sites, where $CO_2$ show flat or multiple peaks in winter (Barlow et al., 2015). Hence, deriving other metrics like the timing of the carbon uptake period (CUP) from

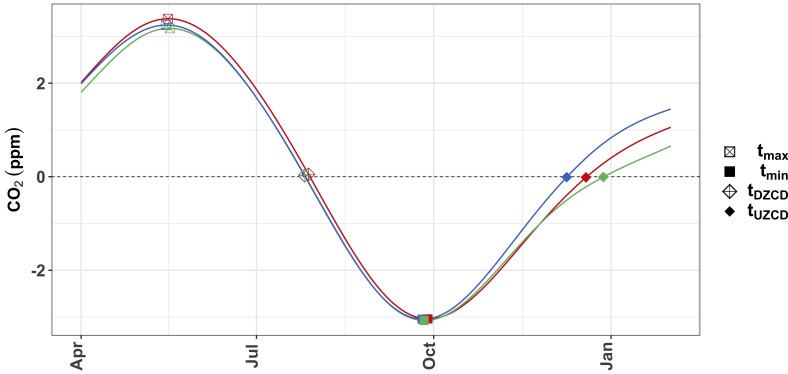

**Figure 1.** Diagram showing how the skewness of the seasonal cycle can influence the estimation of the CUP based on ZCD. The three seasonal cycles have similar seasonal cycle maxima, minima and downward ZCD but very different upward ZCD.

the seasonal cycle maximum results in less robust estimates. The CUP is defined as the time of the year during which the $CO_2$ uptake is greater than the $CO_2$ release. The onset and termination of the CUP are marked by the spring maximum and late summer minimum of the seasonal cycle, respectively. However, the seasonal cycle at many observational sites is characterized by a flat peak or multiple peaks in winter, making it difficult to estimate the start of the CUP. To avoid this problem, previous studies have used the comparatively more unambiguous ZCD to approximate the timing and duration of the CUP (e.g. Piao

et al., 2008). The ZCD are the two dates in the seasonal cycle when the detrended $CO_2$ curve crosses the zero-line (an imaginary line passing through 0 ppm in the detrended $CO_2$ seasonal cycle). Note that this period starts later than the seasonal spring maximum and ends later than the summer minimum, i.e., it is shifted compared to the CUP definition above. This approximation is thus based on the assumption that, if the shape of the seasonal cycle does not change significantly, a change in the phase at one point (e.g., maximum) of the seasonal cycle can be traced as a relative phase change at other points (Barichivich et al.,

2012). However, the shape of the seasonal cycle changes from year to year, and the CUP approximated using the ZCD may be erroneous (Barlow et al., 2015). This is illustrated in Fig. 1.

Barlow et al. (2015) show that using the time-derivative of a time-series can provide a more robust estimate of the key dates that define the CUP, compared to the conventional use of ZCD. A threshold of this time-derivative as a fraction of peak uptake in mid-summer was shown a robust metric to define both start (threshold 25%) and end (threshold 0%) of the CUP in their

study. They used a synthetic data experiment applying a linear trend with substantial interannual variations in amplitude ($\pm$ 25%) and CUP ($\pm$10 days) to a $dCO_2/dt$ time series, to show that in the absence of transport, their method can capture the prescribed linear trend of the CUP. We expand on that work here by additionally creating an ensemble of fitted time-series using residual bootstrapping on a loess-fit. For each ensemble member we calculate the first derivative, allowing us to determine the timing of the various start, end, and peak moments in the CUP, its duration, and the individual uncertainty on each

metric for each individual year in the time series. We call this the ensemble of first derivatives method (EFD method). The EFD method accounts for the random and non-linear changes from year to year in the $CO_2$ time-series, allowing a better handling

**Table 1.** Observational sites of NOAA/ESRL network used in this study

| Station name | Station code | Latitude | Longitude | Time period | Data Source |
|---|---|---|---|---|---|
| Mauna Loa, Hawaii, United States | MLO | 19.47°N | 155.57°W | 1977-2017 | (Dlugokencky et al., 2019) |
| Assekrem, Algeria | ASK | 23.26°N | 5.63°E | 1996-2018 | (Dlugokencky et al., 2020) |
| Sand Island, Midway, United States | MID | 28.21°N | 177.36°W | 1986-2018 | (Dlugokencky et al., 2020) |
| Weizmann Institute of Science at the Arava Institute, Ketura, Israel | WIS | 29.96°N | 35.06°E | 1996-2018 | (Dlugokencky et al., 2020) |
| Terceira Island, Azores, Portugal | AZR | 38.76°N | 27.37°E | 1996-2018 | (Dlugokencky et al., 2020) |
| Niwot Ridge, Colorado, United States | NWR | 40.05°N | 105.58°W | 1976-2018 | (Dlugokencky et al., 2020) |
| Shemya Island, Alaska, United States | SHM | 52.71°N | 174.12°E | 1986-2018 | (Dlugokencky et al., 2020) |
| Barrow Atmospheric Baseline Observatory, United States | BRW | 71.29°N | 156.61°W | 1972-2017 | (Dlugokencky et al., 2019) |
| Ny-Alesund, Svalbard, Norway and Sweden | ZEP | 78.90°N | 11.88°E | 1995-2018 | (Dlugokencky et al., 2020) |
| Alert, Nunavut, Canada | ALT | 82.50°N | 62.50°W | 1986-2017 | (Dlugokencky et al., 2019) |

of outlier years (in mean or uncertainty), which potentially improves trend-analyses of seasonal cycle changes. We apply the EFD method to long time-series and a set of stations covering the low, mid and high latitudes.

We first use the EFD method to confirm that the $CO_2$ ZCD are not the best proxy for determining the timing and duration of the CUP, also when the newly derived uncertainty is considered. We then demonstrate that the EFD method is independent of the skewness of the seasonal cycle, and we optimize the threshold for the CUP onset and termination based on the first derivative. The derived uncertainty also reveals that the robustness of various metrics are site-dependent, with high-latitudes being sensitive to the seasonal cycle maximum (also found in Barlow et al. (2015)), and low latiude sites sensitive to the upward zero-crossing date (UZCD) of the $CO_2$ seasonal cycle. We also tested if the EFD method is sensitive to the specific curve-fitting method applied by fitting the data with the commonly-used CCGCRV method, which is a frequency-domain-based filter, similar to the wavelet transform approach of Barlow et al. (2015). The measurements used in this study are presented in Sect. 2 and the EFD-method is presented in Sect. 3. The results and the discussion on the findings can be found in Sect. 4 and Sect. 5 respectively, and Sect. 6 summarizes the findings of this study.

## 2 Data

We use discrete $CO_2$ dry air mole fraction from flask measurements from ten observational sites of the NOAA/ESRL network (Dlugokencky et al., 2019, 2020), ranging from 19°N to 82°N latitude. Table 1 lists the station names, station codes, their locations and the studied time period for each station (longer time records are available for MLO and NWR but these years have large data gaps of an year or more hence are not considered for analysis). At these observational sites, air is sampled

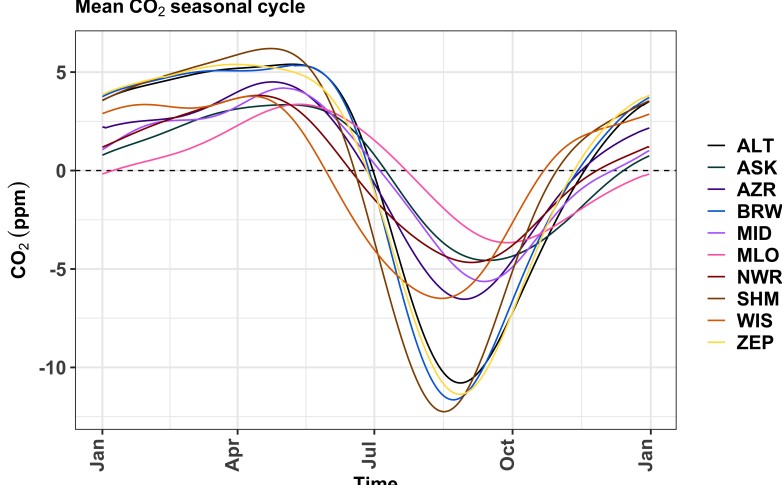

**Figure 2.** Mean seasonal cycle of $CO_2$ at the stations studied. Note that this seasonal cycle is derived from the fitted loess-curves and excludes the observed mean, trend, and high-frequency variations in $CO_2$.

in glass flasks under background conditions, hence the dry air mole fractions from the air samples are representative of the zonal mean atmospheric composition (Langenfelds et al., 2002). These air samples are collected weekly in pairs for quality control, and pairs with a difference less than 0.5 ppm between the two samples are flagged as good-quality data ("good pairs") (Dlugokencky et al., 2019, 2020). For our analysis, we use the mean value of each pair considered as "good pairs" and exclude low-quality measurements, which introduces irregular gaps in the data. The mean seasonal cycle of the higher latitude stations (above 45°N latitude, i.e. SHM, BRW, ZEP, and ALT) is characterized by a broader maxima or multiple peaks in winter. Some lower latitude stations like MLO, MID and NWR have distinct seasonal cycles with clearly defined maxima, while others, like ASK, AZR and WIS, have broader peaks (Fig. 2).

## 3 Method

### 3.1 Loess fitting

The time series of $CO_2$ can be described as the superposition of different modes of variability, acting at different frequencies. A standard approach to extract these modes of variability from the observations ($X_{obs}(t)$) is to define:

$$X_{obs}(t) = X_{trend}(t) + X_{seas}(t) + R(t) \tag{1}$$

where $X_{trend}(t)$ is the low frequency component of the data, which captures variability on multi-annual time scales; $X_{seas}(t)$ represents the seasonal cycle, which can be expressed in terms of a series of harmonics; and R(t) captures the remaining variability (Cleveland et al., 1990). The data used in this study are provided at approximately weekly time steps and includes gaps,

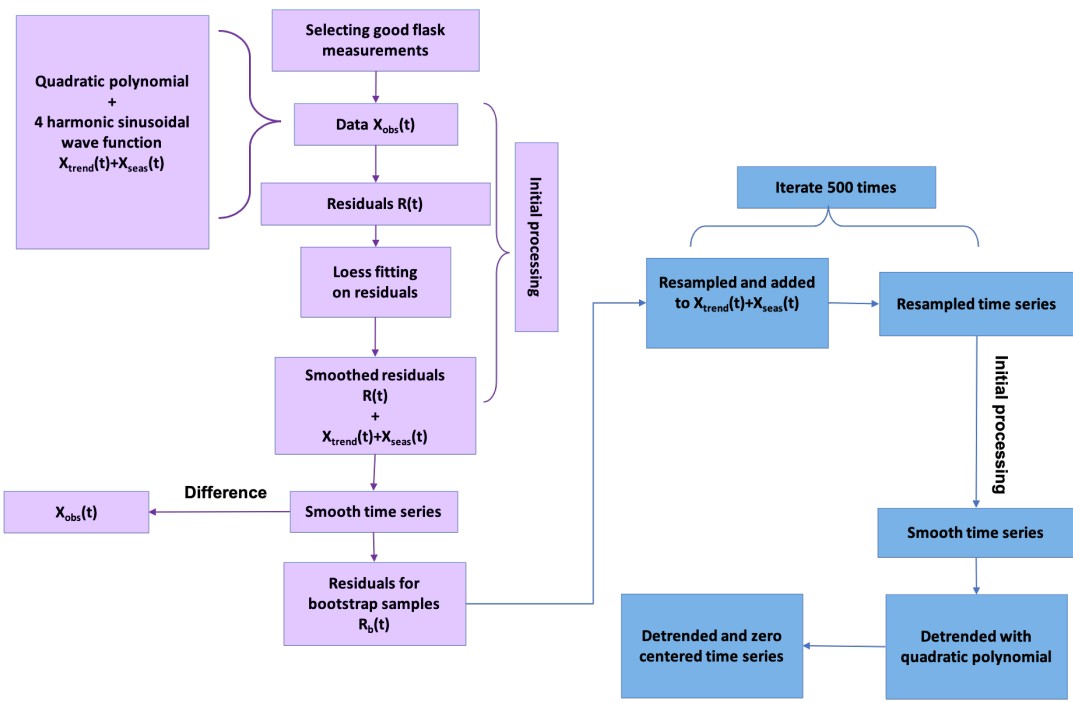

**Figure 3.** Flow diagram explaining the processes of curve fitting (purple boxes) and ensemble generation (blue boxes).

as described above. We fill gaps and estimate daily values by fitting a series of curves described in Eq. (1) and use residual bootstrapping (Kreiss and Lahiri, 2012) to generate an ensemble of 500 fitted curves consistent with the observational data for uncertainty estimation. Figure 3 describes the steps involved in curve-fitting and uncertainty estimation. Each step is described in detail below.

115

First, we separate the long-term trend and mean seasonal cycles ($X_{trend}(t) + X_{seas}(t)$) with a second-degree polynomial and four harmonic sinusoidal functions respectively (Bacastow et al., 1985). The remaining variability, R(t), is referred to as the residuals, which we verified to not show autocorrelation. We then fit a smooth curve to the residuals using the "loess" (local regression) method, which smooths the data, taking into account the gap-lengths in the data. The "Caret" package (Kuhn,

120    2020) in R provides a method for optimizing the smoothing parameter for the "loess" regression using a mathematical method called k- fold cross validation. The optimization is based on five repetitions of ten fold (k=10) cross-validation, where the sub-samples are randomly sampled with restitution. The optimized smoothing parameters are then used to fit a smooth curve to the residuals (R(t)). The resulting smoothed residuals ($R_s(t)$), which contain the remaining variability, are added back to the other components ($X_{trend}(t) + X_{seas}(t)$). This produces a continuous and smooth data set that preserves short-term variations.

## 3.2 CCGCRV fitting

CCGCRV is a curve fitting method developed by Kirk Thoning and Pieter Tans (Global Monitoring Laboratory (GML), NOAA) in the late 1980s. The method fits a combination of polynomials and annual harmonics to the data to approximate the long-term variation and seasonal cycle. The short-term and interannual variability are retained by filtering the residuals from the fit using a low-pass filter. A detailed description of the routines used for fitting the data and filtering of residual can be found in Thoning et al. (1989). In this study we use the C language version of CCGCRV, freely available at: ftp://ftp.cmdl.noaa.gov/pub/john/ccgcrv/ for curve-fitting and finally obtaining a detrended time-series. The values chosen for the input parameters were taken from Table 2 of Pickers and Manning (2015), who optimized them by fitting artificial data (short-term cut-off period $f_s$: 250 days; long-term cut-off period $f_l$: 1500 days; number of harmonic terms: 4; degree of polynomial function: 3).

## 3.3 Ensemble generation

Further, for uncertainty estimation, we generate 500 bootstrap samples from the curve fitted data. For this, we calculate the difference between the smoothed data and the observational data which gives the new set of residuals for generating bootstrap samples. These residuals are resampled (with replacement) and added to the initial fitted curve, producing a resampled time series. The resampled time series is processed as described in the preceding sections to obtain a continuous and smooth data set with daily values. The residual resampling and further processing are iterated 500 times to create an ensemble of 500 slightly different de-trended time series (bootstrap samples) which are all consistent with the observations (Fig 3 shows these steps for loess fitting). The classical bootstrapping method (where the observations are resampled) cannot be applied directly to a time series data as the resampling step fails to replicate the time-dependent structure. Hence, we use residual bootstrapping where bootstrapping is applied to the residuals obtained from fitting a model to the raw data. The resampled time series thus show the same time dependence as the observational data, but are produced from the fitted curve and a random component from the residual resampling.

The ensemble of fitted curves is used to constrain the uncertainty in seasonal cycle metrics estimates. If the estimated metrics differ largely across the bootstrap samples it indicates that the metric estimate is influenced by the inherent uncertainty in extracting a definitive seasonal cycle, by curve fitting the discrete data. Hence, interpreting these metrics without accounting for this uncertainty can be misleading.

## 3.4 Ensemble of first derivative (EFD) method

At high-latitude measuring stations the CUP extends from the seasonal cycle maximum in spring to the seasonal cycle minimum in late summer (Barichivich et al., 2012), driven by $CO_2$ uptake by ecosystems in the Northern Hemisphere. There is large uncertainty in associating the seasonal cycle maximum with the onset of the CUP, and the definition of the maximum is very sensitive to the curve-fitting method (Barichivich et al., 2012). The uncertainty in associating the timing of a maximum to

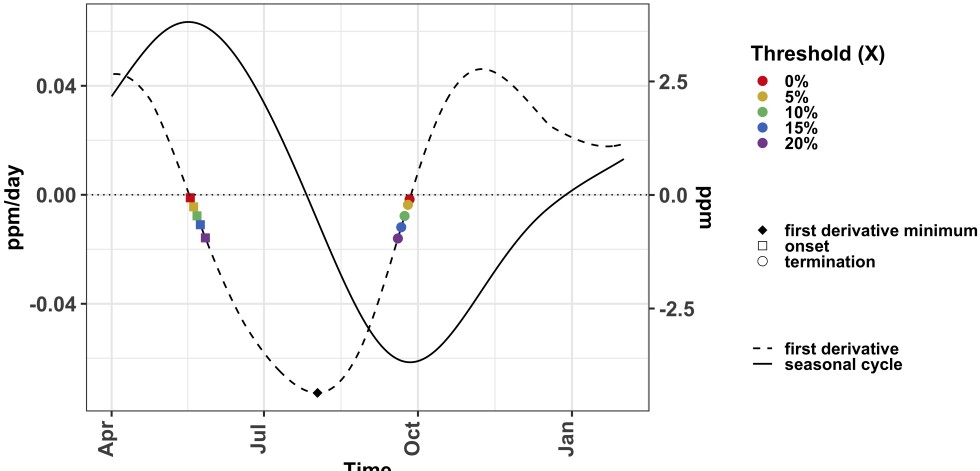

**Figure 4.** Schematic diagram showing the timing of the CUP as determined by the first derivative method. The timing is marked by a threshold, defined in terms of the first derivative of the $CO_2$ seasonal cycle. It is defined as X% of the first derivative minimum. The value of X is varied from 0% to 20% and the corresponding threshold value is marked on the seasonal cycle first derivative with different colored points. Their timing then defines the timing of the CUP for the different threshold values. The day of the onset and the termination of the CUP are defined by the points before and after the first derivative minimum respectively. The squares and circles denote the onset and threshold calculated with different thresholds.

the start of the CUP is larger than associating it with the ZCD, especially if the seasonal cycle is characterized by a fairly flat peak, or multiple peaks during the winter (Piao et al., 2008). Hence, previous studies have used the ZCD and their difference as proxies for the onset, termination and duration of the CUP, respectively. However, the period between the ZCD includes the

160  $CO_2$ release period that does not directly affect the CUP (Fig. 1). Therefore, we use the alternate method proposed by Barlow et al. (2015) to determine the timing and duration of the CUP from the first derivative of the mole fraction data, which more closely corresponds to the spring maximum and the late summer minimum times. We then, estimate the uncertainty in the different CUP estimates by using the spread of the ensemble members.

165  For each ensemble member we calculate the first derivative of the time series as a proxy for the rate of $CO_2$ uptake or release. The first derivative is at its minimum when $CO_2$ uptake is most intense and reaches zero at the peak or trough of the seasonal cycle, i.e. when the sign of the integrated large-scale $CO_2$ flux changes. However, a peak or a trough (as indicated by zero first-derivative Fig. 4) might not correspond to the spring maximum or late summer minimum if the peak is flat or there are multiple peaks in winter. The timing of the CUP should be such that it closely corresponds to the timing of the spring

170  maximum and late summer minimum.

To determine the onset and termination of the CUP from $CO_2$ mole fractions, we define a threshold, based on an ensem-

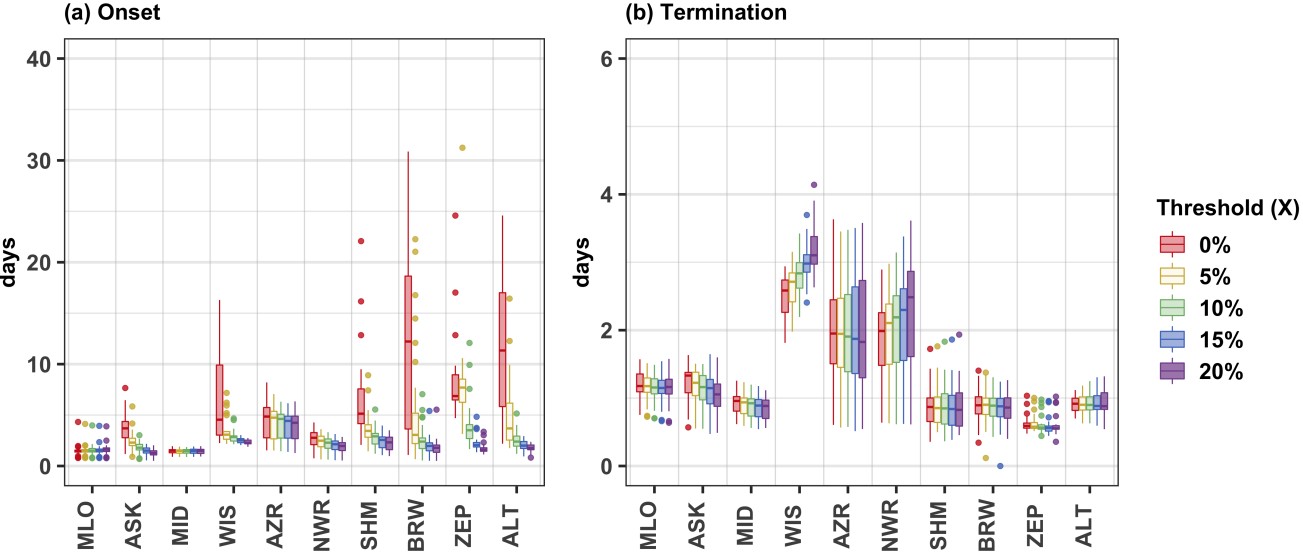

**Figure 5.** Standard deviation (among ensemble members) in the onset (a) and termination (b) of the CUP across years. Differently colored boxes represents different threshold values. The threshold value with X equal to 15% and 0% is chosen for defining the onset and termination of the CUP respectively, for all the studied sites.

ble of the first derivative of the time-series. We define the threshold in terms of the first derivative of the $CO_2$ dry air mole fraction measurements analogous to Barlow et al. (2015). The first derivative can be seen as a proxy for the flux (not an exact correspondence, as the seasonal cycle at each site is affected by the atmospheric transport). The threshold is defined as X% of the first derivative minimum and X is determined separately for the onset and termination of the CUP. The onset/termination of CUP is defined as the closest point to the threshold value before/after the first derivative minimum (Fig. 4). The threshold for the onset and termination is chosen such that 1) the uncertainty in the timing of onset and termination is minimized across the ensemble members and 2) it represents as long a period as possible within the CUP. We varied the value of the parameter X until we find the optimum threshold. When X is 0%, it corresponds to the time period between the seasonal cycle maximum and minimum, including the full CUP but additional non-CUP periods may be erroneously included due to multiple peaks or flat maxima. By increasing the value of X we remove this error, but can also truncate part of the "actual" CUP. Hence, we try to select a low value of X while reducing the uncertainty in the timing of the CUP.

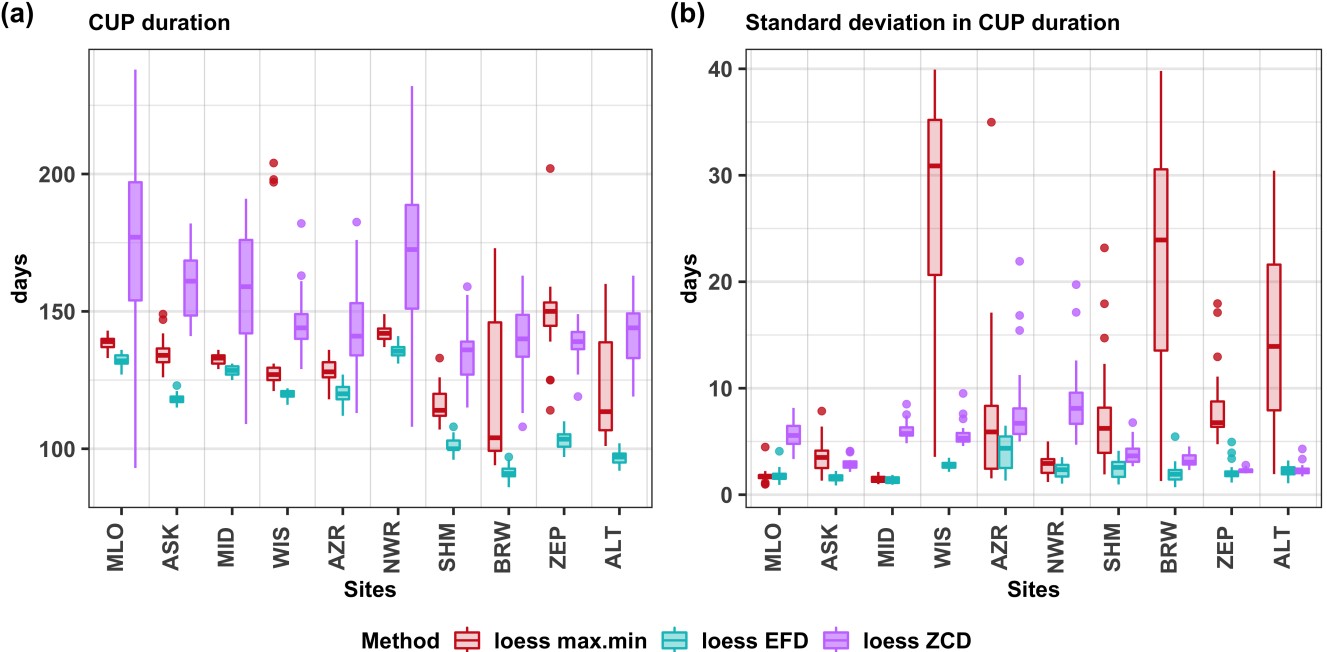

**Figure 6.** Boxplot showing the distribution of the (a) median and (b) standard deviation of CUP duration across all years estimated for the loess-fitted residual bootstrap samples using different methods at the studied sites. The median of the CUP duration for a given year is estimated from the ensemble spread for that year. The CUP duration is calculated using three methods, namely: as the period between the seasonal maximum and minimum (loess max.min), the EFD method (loess EFD), and using the ZCD (loess ZCD)).

## 4   Results

For the EFD method, we first optimize the threshold as described in Sect. 3.4. By continuously increasing X we found the optimum for the termination is 0% and for the onset it is 12-13%, with maximum CUP representation and no further reduction in the uncertainty beyond it. To be on the safe side, we chose 15% as the threshold (for onset) in all our analyses. Incidentally, previous studies using flux measurements have also used 15% of the maximum GPP as a threshold to define the start of the growing season (e.g. Wang et al., 2019). The result from varying X in steps between 0%-20% is shown in Fig 5. When X is set to 0%, the onset corresponds to the maximum of the seasonal cycle, hence the large spread in CUP-onset at BRW and ALT. The interquartile range of 15.0 and 11.2 days respectively can be attributed to the multiple peaks or broad peak of the $CO_2$ seasonal cycle at these stations. Compared to the onset (30.8 days shown as whiskers of the boxplots in Fig.5 (a)) the variability in the termination of the CUP is smaller with a maximum range of 3.9 days (whiskers of the boxplots in Fig.5 (b)). The standard deviation in the termination is highest at WIS where the median of the boxplot for different threshold values is within 2.8 ±0.2 days. Hence to define the termination, we chose a threshold such that the standard deviation is minimized at WIS. This is achieved when X is set to 0% (the median of the spread is then 2.5 days).

We estimate the duration of the CUP for each year using different approaches: 1) the difference between the seasonal cy-
cle maximum and minimum times, 2) the difference in the ZCD and 3) using the EFD method. Figure 6 (a) shows the duration
of CUP for all the studied sites across the years (median of the 500 ensemble members), estimated using the three different
methods. The size (interquartile range) of the boxplots varies strongly across the stations for CUP duration calculated using
the ZCD and the maxima and minima. At the lower latitude stations MLO, MID and NWR, the variability in the CUP duration
is larger than at the other stations when using the ZCD ("loess ZCD" in Fig. 6 (a)). This is seen in the interquartile range of
the "loess ZCD" boxplots, with values of 43, 34 and 37 days for MLO, MID and NWR respectively, which is larger than for
the other stations (within 15.2 ±5.1 days). The large interquartile range of the CUP-duration estimates using maximum and
minimum times at the high latitude sites BRW (46.75 days) and ALT (32 days) ("loess max.min" in Fig. 6 (a)) follows mainly
from the large variability in the timing of the seasonal cycle maximum across the ensemble members (Fig. 7).

When using EFD method, the CUP estimates have least uncertainty across the ensemble members (Table 2). Figure 6 (b)
compares the standard deviation of the CUP duration across years at all studied sites and methods. The standard deviation is
smaller when the EFD method is used for calculating the CUP duration, implying that this metric is better determined. The
interquartile range of standard deviation is largest for the method using the dates of the seasonal cycle maximum and minimum,
especially at higher latitude stations like BRW (17 days) and ALT (13.7 days) and lower latitude station like WIS (17 days). At
MLO, MID and NWR, using the ZCD to approximate the CUP duration results in a larger standard deviation (median of the
spread is 5.5, 6.7 and 8.1 days respectively) in the CUP duration relative to the other methods used (the median of the spread
for the other methods is within 1.75 ±0.6 days).

Here we show that using the EFD method, the uncertainty in the CUP estimate is reduced across all the studied sites. Previous
studies (Barichivich et al., 2012; Barlow et al., 2015) also noted the large uncertainty in using the seasonal cycle maximum
and minimum to determine the CUP, which is similar to our result (Fig. 6 (b)). Therefore this method will not be considered
in further analysis here. However, when ZCD is used to approximate CUP duration there is also large uncertainty at the lower
latitude stations (e.g. the interquartile range for the 'loess ZCD´ boxplot for MLO is 43 days, Fig. 6(a)). Nevertheless, the ZCD
is a widely used approach (e.g. Piao et al., 2008; Barichivich et al., 2012, 2013), therefore the EFD method is compared against
the ZCD here. The difference between the CUP estimates, using the two different methods (EFD and ZCD) varies from year
to year, suggesting that the estimates cannot be corrected by simply adding an offset (Fig. 8). The X-axis range, showing the
CUP from ZCD in Fig. 8, has large year-to-year variation in the CUP, with the largest variation at MLO, NWR and ASK.

To further test the robustness of the CUP estimates based on the loess-fitted residual bootstrap method, we compared them
against the CUP estimates from an ensemble using the CCGCRV curve-fitting method. Comparable results were obtained
when the same CUP estimation method (ZCD / EFD) was applied to the ensemble members using the two different curve-
fitting methods (Fig. 9 (a)). The CUP duration calculated from the CCGCRV ensemble using the ZCD and the EFD method
were within the range of their corresponding estimates from the loess-fitted ensemble members. The mean difference between

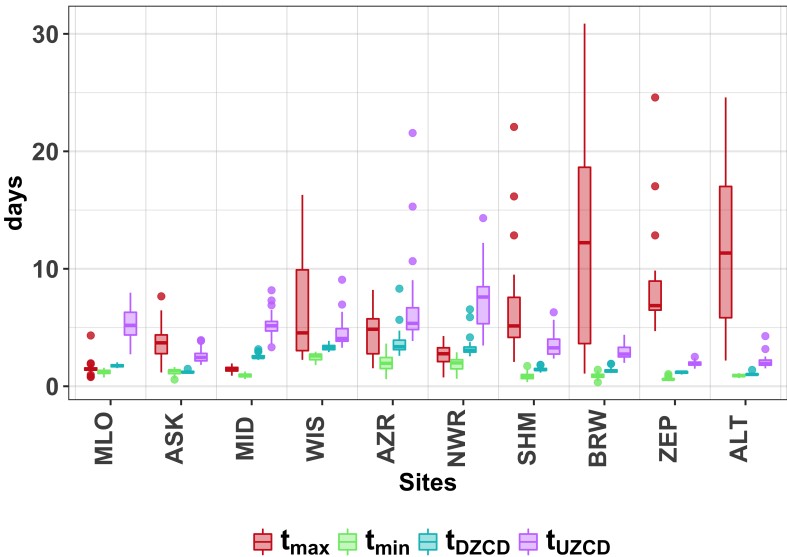

**Figure 7.** Boxplots showing the bootstrap standard deviation (i.e. the standard error of estimate) in the timing of the seasonal cycle maximum ($t_{max}$), minimum ($t_{min}$) and upward ($t_{UZCD}$), downward ($t_{DZCD}$) zero crossing dates across the years for loess fitted residual bootstrap samples. The standard deviation in the timing of the different metrics for a year is estimated from the ensemble spread for the year. In the box-plots in this paper, the box denotes the interquartile range (IQR), showing the median with a solid line. The whiskers range from $Q_1$-1.5x IQR to $Q_3$+1.5x IQR, where $Q_1$ and $Q_3$ are the 25th and 75th percentiles.

the median of the "CCG ZCD" and "loess ZCD" estimates in Fig. 9 (a) is 1.1 days, and between the median of "CCG EFD"
and "loess EFD" estimates the mean difference is only 0.6. However, the range of boxplots corresponding to "loess ZCD" is
larger relative to the "CCG ZCD", resulting from the curve-fitting details. In the loess method the long-term trend in the data
is separated by fitting a quadratic polynomial, the decadal variability in the data is then retained which influences the ZCD
leading to more variability in the CUP approximated using the ZCD. The EFD method is less sensitive to the choice of the
curve-fitting method (Fig. 9 (a)) shown in the comparable "CCG EFD" and "loess EFD" numbers. Furthermore, we show that
for both curve-fitting methods the standard deviation in the CUP duration estimate across the ensemble members is lowest for
EFD method (Fig. 9 (b)). Thus, EFD method produces robust results irrespective of the particular curve-fitting method.

The CUP duration approximated using the ZCD shows larger spread for sites like MLO (with an interquartile range of 16
days for CCGCRV fitted data and 43 days for loess fitted data) irrespective of the curve-fitting method used. This is attributed
to variability in the UZCD due to the skewness of the seasonal cycle during periods of net release and is similar in both the
curve-fitting methods. Furthermore, we find that using the EFD method of CUP estimation resulted in smaller spread across
the bootstrap samples for both the curve-fitting methods (Fig. 4). This suggests that the period within the onset and termination

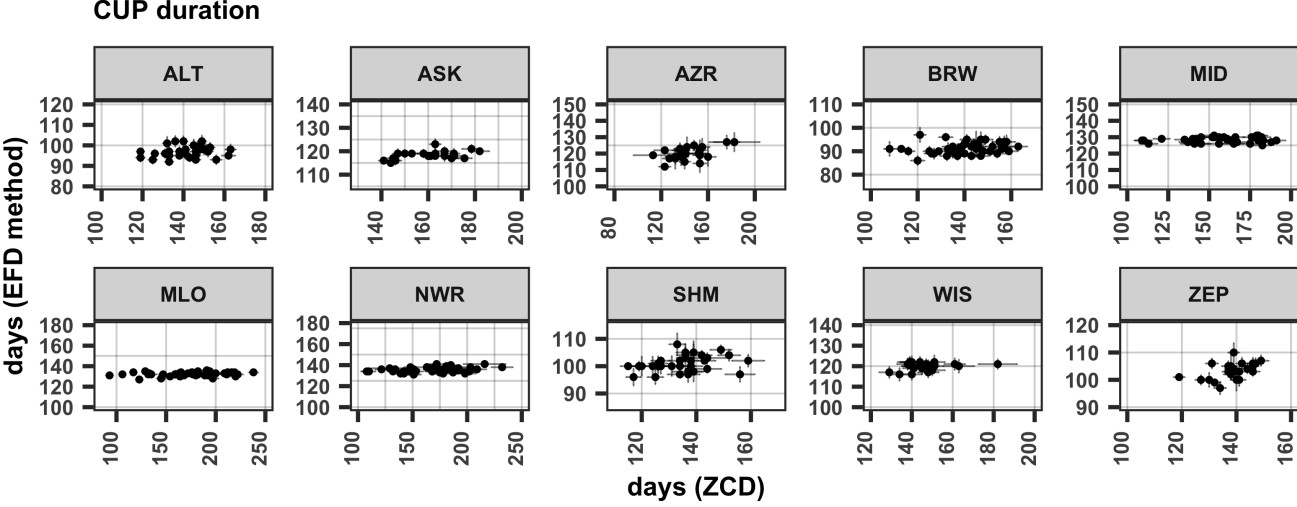

**Figure 8.** CUP duration estimated from the loess-fitted residual bootstrap samples using the timing of the ZCD (x-axis) against that estimated using the EFD method (y-axis) for different sites (panels). The points show estimates for different years and the associated error bars show the spread (median ±sd) of the ensemble.

defined by the EFD method, which includes only part of the drawdown period, is less variable than the time period between the ZCD, which includes parts of both the drawdown and release periods.

## 5  Discussion

### 5.1  Uncertainty estimation with EFD method

In this study, we quantify the uncertainty in the $CO_2$ seasonal cycle curve-fitting by creating multiple residual bootstrap samples. The spread of this ensemble provides a measure of the uncertainty in the estimation of seasonal cycle metrics. The ensemble members are consistent with the observational data; hence we consider the variability of the metric estimate across the ensemble as a measure of uncertainty. The ensemble approach allows us to quantify the year-to-year change in different seasonal cycle metrics and we see that the sensitivity of these metrics to curve-fitting differs across latitudes and from year to year. Here we show that $CO_2$ seasonal cycle metric estimates can be strongly sensitive to the method used, hence any method must be thoroughly evaluated before it can be used to derive trends from the atmospheric data. In Barlow et al. (2015) the robustness of the first derivative is tested by evaluating its ability to capture a known trend from a synthetic time series. They found a larger threshold value for the onset (25%, suggesting a shorter CUP in their approach) from a synthetic data trend

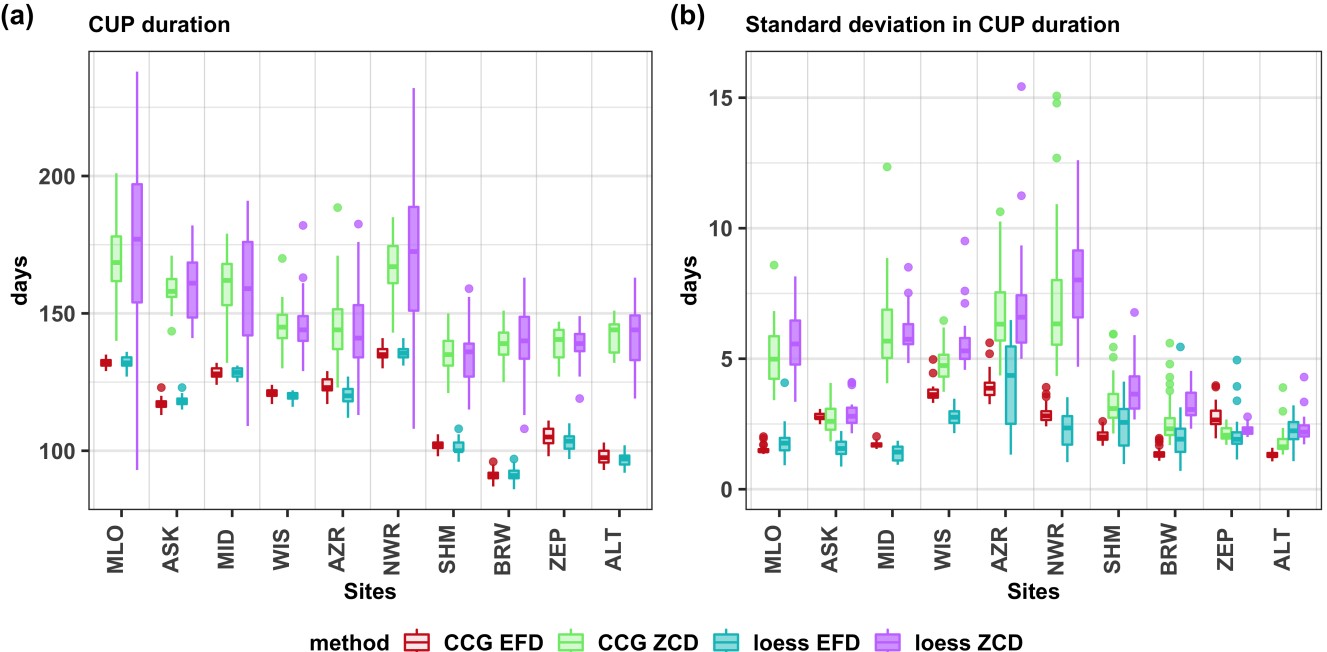

**Figure 9.** Boxplot showing the distribution of the (a) median, (b) standard deviation of the CUP duration across years (as described in Fig. 6), estimated for the CCGCRV-fitted residual bootstrap samples using the EFD method (CCG EFD, red), the ZCD (CCG ZCD, green), and for the loess-fitted residual bootstrap samples using the EFD method (loess EFD, blue) and the ZCD(loess ZCD, purple).Note that six outliers (with values between 16 to 30 days) corresponding to AZR in (b) has been removed for ease of visualization.

analysis in which they applied a linear trend with Gaussian variations of the peak uptake date to a $CO_2$ time series. However, we argue that the data-derived year-to-year uncertainty from our ensemble provides a more robust threshold estimate and we derived a tighter threshold than Barlow et al. (2015) (15% for onset). Further, Barlow et al. (2015) showed that their method can retrieve the true linear trend to within 10-25%. Our EFD approach provides uncertainty on the year to year variability in
the seasonal cycle metrics based on the fitted data residuals, which could be used in a trend analysis to give differential weights to each year. Also, trend analysis on the individual ensemble members would allow uncertainty on the trend to be calculated. Our demonstration of the EFD method on the CUP could be extended to other metrics that are derived directly from the seasonal cycle in a similar way, for example the peak to trough amplitude, especially when curve-fitting discrete data, at sites with broader or multiple peaks. In a similar fashion, the ensemble-based approach could be used to evaluate a newly proposed
method or select an optimal method for evaluating any other metric based on reduced uncertainty. The EFD approach, based on residual bootstrapping, assumes that the residuals are independent, and as such may not be appropriate for hourly or daily in-situ data. There is considerable auto-correlation between consecutive days in daily measurements, limiting the number of independent events to five or six per month, which is comparable to the scale of our weekly measurements. The uncertainties in curve-fitting related to gaps in the data are also less in the case of more frequent measurements.

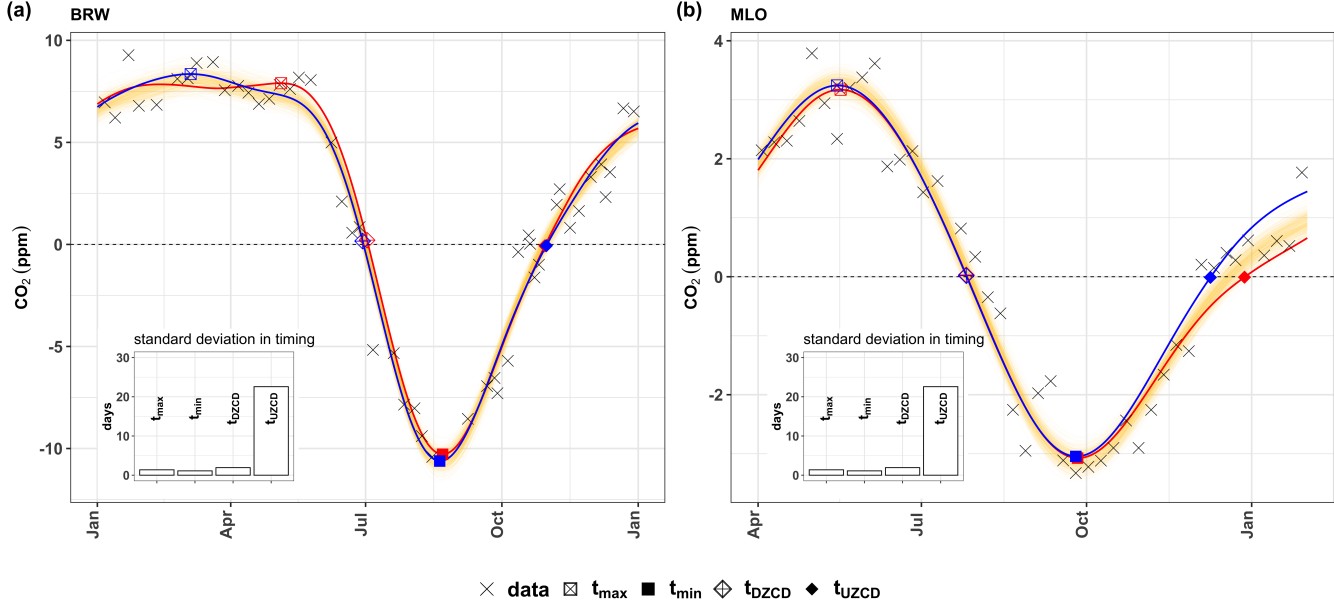

**Figure 10.** Fitted bootstrap samples (thin yellow lines) representing the seasonal cycle of a year at (a) BRW and (b) MLO. The observational data of the corresponding time period are shown with '×'. Red and blue curves in (a) and (b) highlight two random ensemble members that differ in shape and are marked with the timings of the seasonal cycle maximum ($t_{max}$), minimum ($t_{min}$), downward zero-crossing($t_{DZCD}$) and upward zero-crossing ($t_{UZCD}$) with the corresponding symbols as in the legend. The vertical bars in the inset show the standard deviation of the labeled metric estimates across the ensemble members.

## 5.2 Latitudinal dependence of metrics

The shape of the $CO_2$ seasonal cycle varies with latitude. At the higher latitude stations the seasonal cycle has a broader peak or multiple peaks in winter and the timing of the seasonal cycle maximum cannot be interpreted as the onset of the CUP. Further, our analysis shows that there is large uncertainty in the timing of the seasonal cycle maximum (Fig. 7) at higher latitude stations which is in agreement with previous studies (e.g. Barichivich et al., 2012; Piao et al., 2008; Barlow et al., 2015). For example for BRW shown in Fig. 10 (a) the atmospheric mixing ratios have a nearly constant value from January to May followed by a decrease in $CO_2$ until a minimum is reached in August, also illustrated in Barlow et al. (2015) (their Figure 6). If the seasonal cycle were determined solely by the biospheric fluxes then the onset and termination of the CUP would be marked by the timing of the seasonal cycle maximum and minimum respectively (Barichivich et al., 2012). However, it can be noted that the estimated timing of the seasonal cycle maximum varies greatly across the bootstrap samples in BRW (inset of Fig. 10 (a)), where the seasonal cycle is characterized by a flat peak. An earlier peak is likely to be associated with a flat maximum or multiple peaks that may result from transport (Parazoo et al., 2008) or other fluxes, rather than indicating the onset of the uptake period (Barlow et al., 2015). The timing of the ZCD at BRW are consistent across the ensemble members, which suggests that the timing of the ZCD (upward and downward) is less ambiguous. Other higher latitude sites like ALT, SHM and

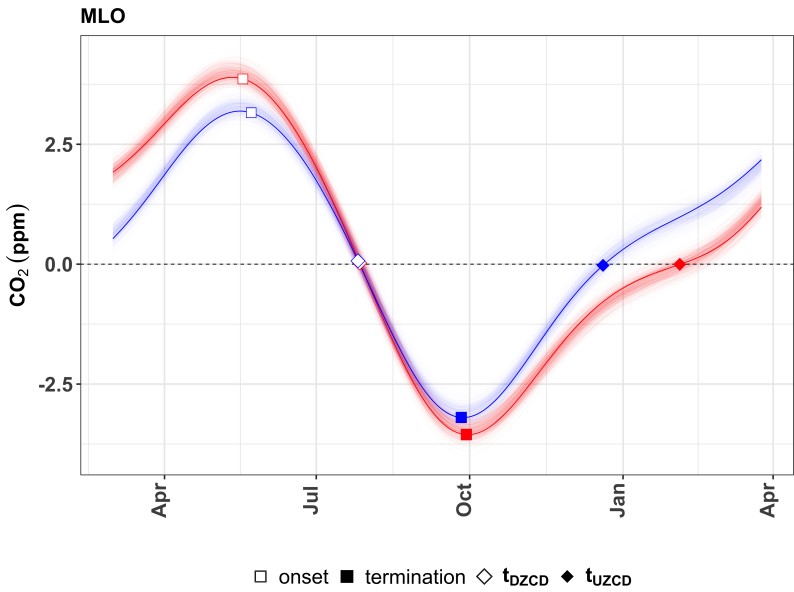

**Figure 11.** Bootstrap samples representing the seasonal cycle of two years (red and blue) with largely different CUP timing when estimated with the two methods (a case taken from Fig 8). Thin red and blue lines represent the ensemble spread for the two years. The thicker red/blue lines represent a random ensemble member from each year and these are marked with the timings of the onset and termination as determined by the EFD method (squares) and the ZCD (diamonds).

ZEP and lower latitude sites like ASK, WIZ and AZR exhibit similar seasonal cycles, characterized by flat or multiple peaks
and less ambiguous ZCD. However, the ZCD are closer to the timing of the maximum uptake and release of $CO_2$ (Manning, 1993) than to the actual onset and termination of the CUP. For example, in Fig. 1, the onset of the CUP occurs in June, however the downward zero-crossing occurs in early July, thus the CUP approximated using the ZCD explicitly excludes the start of the drawdown period.

At lower latitude stations like MLO, MID and NWR, we find that the ZCD can vary across ensemble members as shown in Fig. 7 and in such cases the ZCD are clearly not the best proxy for estimating the duration of the CUP. This is especially the case for the time series at MLO (Fig. 10 (b)), which shows relatively a large spread of 5 days (median of the ensemble spread, rounded to the nearest integer) in the timing of the UZCD across the ensemble members. The seasonal cycle at MLO has well-defined peaks and troughs, hence the timings of the seasonal cycle maximum and minimum show only a small spread
of 1 day (median of the ensemble spread, rounded to the nearest integer) across the bootstrap samples (inset of Fig. 10 (b)). In this case, the EFD method gives a more robust estimate of the CUP duration.

We find that in addition to having a larger annual uncertainty, the range of CUP values over the study period for the ZCD approach is much larger than that of the EFD approach for some sites (Fig. 8). For example, at MLO the zero-crossing-

approximated CUP ranges from 100 to 250 days, corresponding to a period of 3-8 months. Changes in the length of the growing season in the Northern Hemisphere are not expected to be this large. As an example, Jeong et al. (2011) estimated the length of the growing season using satellite measurements of normalized difference vegetation index (NDVI). When integrating over the temperate northern hemisphere, the length of the phenology-derived growing season was found to vary by less than 25 days from 1982-2008. The ZCD approach includes changes in both the latter part of the net uptake period and the

early release period, making it difficult to separate the contribution of the net uptake and net release periods to the changes in the CUP estimate. To understand this large spread in CUP, we compare two years with very different CUP values estimated by the ZCD at MLO, 1992 with 192 days and 1998 with 147 days (Fig 11). We find that the difference in the CUP estimate is due to the change in the early release period, whereas the uptake periods are essentially the same. When using the EFD method, by contrast, the two years show similar CUP, 134 and 126 days, respectively. By definition, the EFD is not affected by differences

in the net release period, and therefore provides more robust CUP duration estimates.

## 5.3 Sensitivity of different methods to changes in the CUP

To test the sensitivity of the different methods to changes in the CUP, we performed a synthetic data experiment. We compare the EFD and ZCD methods, and additionally an approach where we determine the CUP using a 25% threshold as in

Barlow et al. (2015) but using a single loess curve fitting rather than an ensemble. We manipulated the posterior net ecosystem exchange (NEE) flux by imposing pre-defined CUP changes ($\Delta$CUP_NEE) for land pixels in the Northern Hemisphere that have a clearly-defined seasonal cycle, using a randomly selected baseline year from the Jena Carboscope $CO_2$ Inversion (Rödenbeck et al., 2003) (version ID: sEXT_ocNEET_v2021). The manipulated fluxes were then transported forward using an atmospheric transport model (TM3, Heimann and Körner (2003)). We then sampled the atmospheric mixing ratios at time

steps corresponding to flask data measurements at the station locations. Then, we evaluated the CUP changes deduced from the simulated $CO_2$ mixing ratio ($\Delta$CUP_MR) in response to the imposed $\Delta$CUP_NEE with the three different CUP estimation methods. The experiment shows that the EFD method is more sensitive to the changes in CUP_NEE compared to the ZCD method, which is seen by the higher slope of $\Delta$CUP_MR to $\Delta$CUP_NEE (0.312 vs 0.08, respectively). Further, the uncertainty in sensitivity of the ZCD and EFD methods were estimated by calculating the 95% confidence intervals of the slopes of the

regression lines by fitting multiple (10 000) sets of randomly resampled CUP_MR values (from its distribution) corresponding to each CUP_NEE value (shown in Fig. 12 for MLO). The confidence interval is large for the ZCD method, from -0.195 to 0.355. This implies that $\Delta$CUP_NEE cannot be reliably inferred from $\Delta$CUP_MR using the ZCD.Further, we observed that the approach in Barlow et al. (2015) is similarly sensitive to changes in CUP_NEE as the ensemble-based approach, however the uncertainty in its sensitivity cannot be calculated as only a single CUP_MR value is available for every CUP_NEE, clearly

showing the advantage of the ensemble approach to constrain uncertainties in metrics.

As seen from the experiment, the ZCD is the least sensitive to changes in the flux. Hence, years with extreme CUP approximated by the ZCD as in Fig. 8 may not indicate actual changes in the CUP but rather other factors, like reduced net

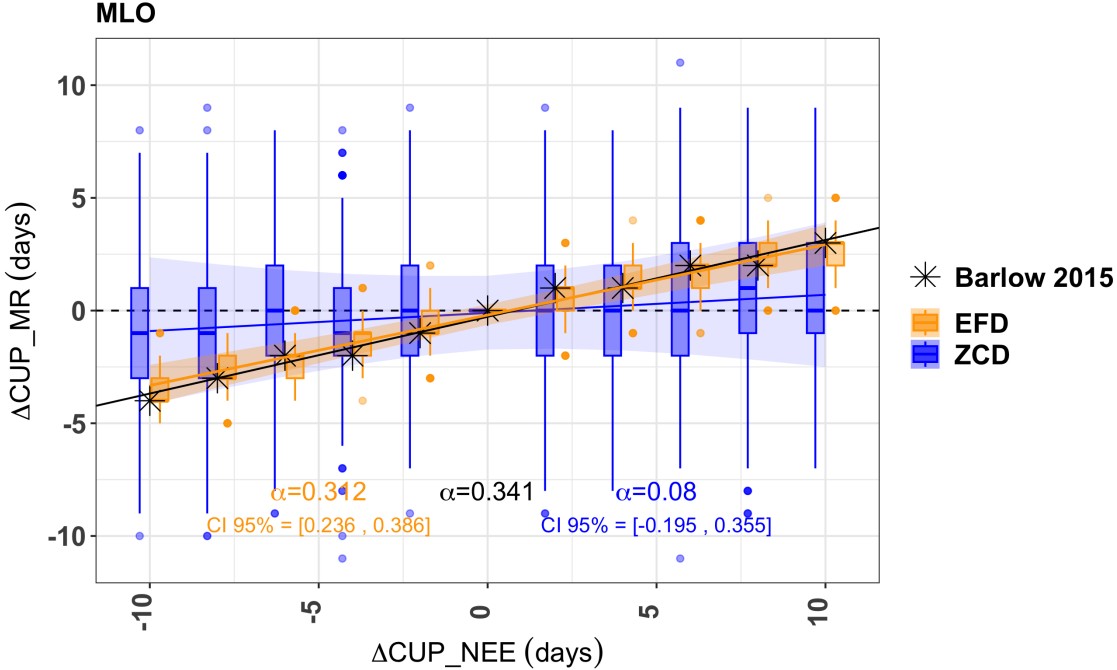

**Figure 12.** Sensitivity of the EFD (orange), ZCD (blue) and an approach similar to that of Barlow et al. (2015) (black asterisks) to imposed changes in the CUP_NEE. The deduced change in the CUP based on the simulated mixing ratio (ΔCUP_MR) is shown here for MLO. The spread of the boxplot for the EFD and ZCD methods corresponds to the uncertainty in the curve fitting estimated by the residual ensemble. The uncertainty in the sensitivity of the EFD and ZCD methods is estimated as described in Sect. 5.3 with multiple regression fits. The shaded region represents the 95% confidence interval (CI) of the fitted lines. The mean slope ($\alpha$) of the fitted regression lines and their 95% CI is shown.

respiration in the early release period, which postpone the UZCD. This shift could be due to reduced net respiration in the
early release period, prolonging the time to reach the UZCD. This change is determined by the interplay of the $CO_2$ uptake and
release processes, which are influenced by physical factors like temperature, soil moisture and solar radiation. For example, in
dry conditions there is less respiration by plants and slower decomposition of organic matter in the soil, resulting in reduced
$CO_2$ release to the atmosphere (Yan et al., 2018). The rate of decomposition further depends on the snow cover and available
detritus content in the soil following leaf senescence. Furthermore, in the early release period, when the solar radiation is not
limiting, plants may continue photosynthesize depending on water availability and temperature, leading to reduced net $CO_2$
release. Thus, in years with extreme CUP as approximated by the ZCD, the physical processes that affects the release period
should be investigated. In comparison to the CUP definition, the approximation by ZCD is also sensitive to variations after the
summer minimum, i.e. during the early release period.

The EFD and the approach in Barlow et al. (2015), which consider the first derivative of the concentration time series as

a proxy for the large-scale spatially integrated flux (Barlow et al., 2015), are relatively more sensitive to the flux changes. However, the inferred changes cannot be directly translated to the underlying flux fields. The concentration time series do not exactly replicate the changes imposed in CUP_NEE, as these changes do not occur simultaneously across the Northern Hemisphere, and hence do not represent a single changed flux, but a superposition of many. These signals dampen each other when aggregated in CUP_MR, as seen in Fig. 12.

The synthetic experiment presented here used simulated data from only one year, thus controlling for the influence of inter-annual variability in atmospheric transport. In reality, atmospheric transport plays an important role in explaining a significant portion of observed $CO_2$ variations at surface stations (e.g. Krol et al., 2018; Fu et al., 2015), which might affect any interpretation of the CUP metrics. An extensive study was carried out by Lintner et al. (2006), confirming the importance of atmospheric transport to account for some of the inter-annual variations in $CO_2$ observed at Mauna Loa. Murayama et al. (2007) showed how year-to-year changes in the atmospheric transport create significant inter-annual variations in the downward zero-crossing date of the $CO_2$ seasonal cycle that cannot be neglected. The ZCD is influenced by transport variability in both the late uptake and early release periods. Hence, changes in the early release period could be erroneously interpreted as changes in the CUP when using the ZCD. Also, atmospheric transport can contribute to the inter-annual variability in CUP estimates when using the EFD method. Hence, we recommend that the contribution of atmospheric transport at the studied background sites should be evaluated before interpreting and relating the CUP metrics to sources/sinks.

## 6 Conclusions

Here, we discuss a method for estimating the timing, duration, and uncertainty of the CUP and related metrics from a discrete time series of $CO_2$ dry air mole fraction data. The uncertainty in the metrics is quantified using an ensemble of fitted time series generated through residual bootstrap sampling, a novel addition to the method presented in Barlow et al. (2015). Previous studies have used the timing of the ZCD as a proxy for defining the CUP, however the timing of the UZCD is influenced by the shape of the seasonal cycle, leading to large variability in the estimated CUP duration across the ensemble members for a given year for some of the studied sites, particularly at lower latitudes. The spread in the CUP duration across the ensemble members for a given year (i.e., the annual uncertainty) is lower for all studied sites when calculated using the EFD method. The EFD method depends directly on the timing and rate of the maximum $CO_2$ uptake; hence the method is not affected by the shape change of the seasonal cycle outside the time period during which the $CO_2$ uptake is larger than the $CO_2$ release. Further, a synthetic data experiment showed that the approaches based on the first-derivative of the $CO_2$ mixing ratio are more sensitive to changes in CUP evaluated at MLO than the ZCD method. With the EFD method the onset and termination is tightly constrained by considering the year-to-year change in the seasonal cycle. To test the impact of the curve-fitting method used, we generated bootstrap samples using both loess-fitted residuals and CCGCRV. The CUP duration estimated using the EFD method results in smaller spread for both curve-fitting methods. Further, for both curve-fitting methods, the standard deviation in the estimates across the ensemble members is smaller when using the EFD method, suggesting that the EFD method gives

robust estimates. Thus, the EFD method allows for a robust estimate of the CUP that better reflects the $CO_2$ drawdown period.

385 This approach could be extended to other metrics of seasonal cycle analysis or to other curve-fitting methods, as was shown with the comparison to the CCGCRV results.

*Data availability.* The $CO_2$ dry air mole fraction data for ALT, BRW and MLO is available from (Dlugokencky et al., 2019) and for the other stations used in this study from (Dlugokencky et al., 2020).

*Author contributions.* The coding and analysis was performed by TK with contributions of MR. The study was conceptualised by JM, AB

390 and MR with contributions from WP. JM, AB, WP, MR and PT contributed with expert knowledge.The original manuscript was drafted by TK which was reviewed and edited by WP, JM, AB, MR and PT.

*Competing interests.* The authors declare that they have no conflict of interest.

**Table 2.** The inter-quartile range of the standard deviation in the CUP duration across all years as described in Fig. 6 (b), rounded to the nearest integer (day). Values are given for three different method of estimation for each site.

| Sites | Time period | sd CUP duration (days) | Method |
|-------|-------------|------------------------|--------|
| MLO | 1977-2017 | 1-2 | max.min |
|  |  | 1-2 | EFD method |
|  |  | 5-6 | ZCD |
| ASK | 1996-2018 | 3-4 | max.min |
|  |  | 1-2 | EFD method |
|  |  | 2-3 | ZCD |
| MID | 1986-2018 | 1-2 | max.min |
|  |  | 1-2 | EFD method |
|  |  | 5-6 | ZCD |
| WIS | 1996-2018 | 21-35 | max.min |
|  |  | 2-3 | EFD method |
|  |  | 5-6 | ZCD |
| AZR | 1996-2018 | 2-8 | max.min |
|  |  | 3-5 | EFD method |
|  |  | 6-8 | ZCD |
| NWR | 1976-2018 | 2-3 | max.min |
|  |  | 2-3 | EFD method |
|  |  | 7-10 | ZCD |
| SHM | 1986-2018 | 4-8 | max.min |
|  |  | 2-3 | EFD method |
|  |  | 3-4 | ZCD |
| BRW | 1972-2017 | 14-31 | max.min |
|  |  | 1-2 | EFD method |
|  |  | 3-4 | ZCD |
| ZEP | 1995-2018 | 6-9 | max.min |
|  |  | 1-2 | EFD method |
|  |  | 1-2 | ZCD |
| ALT | 1986-2017 | 8-22 | max.min |
|  |  | 2-3 | EFD method |
|  |  | 1-2 | ZCD |

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
