# Peer review of "Reducing errors on estimates of the carbon uptake period based on time series of atmospheric CO2"

_Atmospheric Measurement Techniques, 2022_

## Author Comment (AC1)

Response to the reviews of manuscript amt-2022-179: "Reducing errors on estimates of the carbon uptake period based on time series of atmospheric $CO_2$ " by Theertha Kariyathan, Wouter Peters, Julia Marshall, Ana Bastos, Pieter Tans, and Markus Reichstein to Atmos. Meas. Tech.

Questions from the reviewers are written in blue, our answers in black, text copied from the manuscript is written in *italic*, and all changes in the manuscript are typed in red. When referencing page and line numbers, we are always referring to the old version of the manuscript.

During the review process we came across a study by Barlow et al., 2015, where the CUP is estimated using the first derivative approach. Although we developed our approach independently, we no longer claim the novelty of this approach. We rather emphasize the ensemble approach for uncertainty estimation and rename our method to EFD (ensemble of first-derivative method) and include an extensive discussion of Barlow et al., 2015 in our manuscript. The main changes are made to the introduction (line 65) and discussion (line 270).

Answers to Reviewer 1

R1.1
This study presents a novel method to estimate the carbon uptake period (CUP) from discrete $CO_2$ observation time series. The process of determining CUP from discrete time series includes two critical steps: curve fitting and CUP onset and end determination. Curve-fitting methods are needed to interpolate observation at gaps and to filter out the noise and undesirable modes of variability. When analyzing $CO_2$ mole fraction from background observation sites, this means removing the effects of local fluxes or synoptic scale transport variations. Previous studies have shown that the conclusions from the analysis of $CO_2$ time series are sensitive to the choice of the curve-fitting method. CUP estimates are also sensitive to the method used. Previous studies have proposed several methods that use the zero-crossing points or crest and trough of the detrended, zero-centered seasonal cycle. The study presents a new CUP estimation method and provides a detailed uncertainty assessment of the curve-fitting methods and compares them with other methods reported in the literature. The CUP method and the detailed uncertainty analysis of the different curve-fitting methods presented in this study are very relevant. Overall, the paper is well-written and the figures are clear. I recommend the publication of the paper after the following issues have been addressed.

We thank the reviewer for this positive and constructive review. Below, we answer all comments in detail and show the changes that we think have improved the manuscript.

R1.2
It is unclear which methods described in this paper are novel. The FDT method is new and innovative, however, I have reservations about the newness of the rest of the methods. In the abstract, the authors write "…a novel curve fitting method….". The essence of both CCG and the loess method presented here is the same, Equation 1. Is the novel part of the loess method using local regression to smoothen the residuals instead of a low pass FFT filter used in CCG? Or is it that the author's method uses a 2-degree polynomial and 4 harmonic functions while the CCG method uses a 3-degree polynomial and 4 harmonic function? Moreover, the study note that there is no difference in the performance of the loess and the CCG methods (line 254). Could the authors explain what is then the advantage of the proposed loess method? In the rest of the manuscript, the authors only claim the uncertainty generation and FDT methods are new (Line 64, 264 & 323). The ensemble-based method uses bootstrapping to evaluate the uncertainties of a metric. This is again not so new in my opinion. The main novel method presented in this study is the FDT method. I suggest that the authors (1) make clear which methods are novel. (2) restructure the method section so it does not over-emphasize the newness of the loess method.

Thank you for pointing this out. We agree with the reviewer, and have removed claims regarding the novelty of any of the methods used here. We emphasise now how the ensemble-based approach can improve uncertainty estimation by considering the year-to-year changes in the seasonal cycle.

R1.2 (continuation)
(3) if the ensemble-generation method is the same for the CCG and loess methods, describe the ensemble-generation in a separate subsection.

We agree, thank you for the suggestion. The method section has been modified as follows:

1) Added a new subsection 3.3 "Ensemble generation".
2) Moved subsection (*CCGCRV fitting and ensemble generation*) before subsection 3.3 with the following modifications:
      i) Name of section changed from *CCGCRV fitting and ensemble generation* to CCGCRV fitting.
      ii) Lines 166-171 have been removed ("*Further, we generate 500.....).* .
3) this is followed by subsection 3.4, "Ensemble of first-derivative method", where we describe the EFD method- and note the difference from Barlow et al., 2015.

R1.3
The authors have made a good attempt to describe the FDT method. However, I found it difficult to understand how CUP is calculated using the X% threshold. This statement is confusing: "The value of X is chosen to minimize the threshold value (as the rate of uptake towards the beginning and end of the CUP approaches zero) while keeping the uncertainty in timing across the ensemble members small". Does the authors mean the uncertainties are calculated as a function of threshold within the range of 0 to 20 percent, and the onset and termination times are the threshold points where CUP uncertainty is smallest? This becomes clearer in the results section but it will be good to move some of the explanation from the results section to the method section. Perhaps, a figure or an additional panel in figure 4 illustrating this would make the method easier to understand. I also have some concerns about the tested threshold values. Why only 4 discrete values of the threshold were tested? One can easily do this analysis over a continuum. Where does the choice of 0 to 20 percent come from? Why the range does not include positive threshold values, for example, something like -20 to 20 percent?

Thank you for pointing this out. We have followed up this suggestion with an analysis of the threshold over a continuum as suggested, and it is shown and discussed below.

For clarification: the first derivative threshold is determined separately for the onset and termination of the CUP. The threshold should be such that the uncertainty in the timing of the CUP (onset and termination) should be minimized. However, we also want the threshold to capture as much of the CUP as possible. Hence, an optimum threshold should offer a balance between the two requirements. If the seasonal cycle were regulated only by biospheric fluxes, then the CUP could be defined simply by the seasonal cycle maximum and minimum. However, higher latitude sites often have flat or multiple peaks, which leads to ambiguity in determining the onset of the CUP. Therefore, we need a metric that captures the CUP without being affected by the ambiguous timing of the peak. This metric uses the percentage of the first derivative (slope) defined by X.

When X is set to zero, the CUP then corresponds to the time period between the seasonal cycle maximum and minimum. By increasing X continuously to 25 for both the onset and end of the CUP, we see only a smaller fraction of this time period. By progressively increasing X, we truncate more of the drawdown period of the CUP, but we also avoid the ambiguity of the onset timing for the sites with flat peaks. We progressively increased the value of X from zero and found that there was no significant change in the

uncertainty of the CUP timing mostly beyond 12-13% (Figure 1 , blue boxes and beyond). To be on the safe side, we chose 15% as the threshold. Incidentally, previous studies using flux measurements have also used 15% of the maximum GPP as a threshold to define the start of the growing season (e.g. Wang et al., 2019). For clarity, only values of X from 0 to 20 are shown in the manuscript. Negative X values corresponds to points before the maximum and after the minimum of the seasonal cycle, which are outside the time period of interest.

[Figure]

*Figure 1. Similar to Fig 7 (b) in manuscript, tested for threshold over a continuum ( X from 0% to 25%).*

As noted by the reviewer, some of this explanation could be found in the result section, but we will modify subsection 3.3 and Figure 4 as suggested, so that the method is clear. The revised text reads as follows:

Lines 151 is replaced with: The threshold is defined as X% of the first derivative minimum and X is determined separately for the onset and termination of the CUP. The onset/termination of CUP is defined as the closest point to the threshold value before/after the first derivative minimum (Fig. 4). The threshold for the onset and termination is chosen such that 1) the uncertainty in the timing of onset and termination is minimized across 180 the ensemble members and 2) it represents as long a period as possible within the CUP. We varied the value of the parameter X until we found the optimum threshold. When X is 0%, it corresponds to the time period between the seasonal cycle maximum and minimum, including the full CUP but additional non-CUP periods may be erroneously included due to multiple peaks or flat maxima. By increasing the value of X we remove this error, but also truncate part of the "actual" CUP. Hence, we try to select a low value of X while reducing the uncertainty in the timing of the CUP.

Lines 173 added (section 4 Results begin like): For the EFD method, we first optimize the threshold as described in section 3.4. Continuously increasing X we found the optimum for the termination is 0%, and onset is at a value of 12-13%, with maximum CUP representation and no further reduction in the uncertainty beyond it. We then chose 15% as a conservative threshold (for onset) in all our analyses. Barlow et al. (2015) derived a larger threshold value for the onset (25%, resulting in a shorter CUP in their approach) from a synthetic data trend analysis in which they applied a linear trend with Gaussian variations of the peak uptake date to a $CO_2$ time series. We argue, however, that the data-derived year-to-year uncertainty

from our ensemble provides a more robust threshold estimate. The result from varying X in steps between 0%-20% is shown in Fig 5.

[Figure]

*Figure 2. Schematic diagram showing the timing of the CUP as determined by the EFD method. The timing is marked by a threshold, defined in terms of the first derivative of the $CO_2$ seasonal cycle. It is defined as X% of the first derivative minimum. The value of X is varied from 0% to 20% and the corresponding threshold value is marked on the seasonal cycle first derivative with different colored points. Their timing then defines the timing of the CUP for the different threshold values. The day of the onset and the termination of the CUP are defined by the points before and after the first derivative minimum respectively. The squares and circles denote the onset and threshold calculated with different thresholds.*

R1.4

The study focuses on the importance of uncertainties in CUP estimates of the Northern Hemisphere $CO_2$ emissions when estimated using discrete measurements from select background sites. There are intra-annual variations and long-term trends in atmospheric transport which would affect the relationship between the seasonal cycle of the $CO_2$ observations vs the actual emissions (see Krol et al., 2018, Fu et al., 2015). The transport errors will not be an issue when the FDT is applied to a discrete fluxes time series. I suggest the authors add a discussion about the transport-variation-related errors when analyzing fluxes using remote background observation sites to the discussion section.

This is a very important point, thank you for mentioning it. We have now added the following lines in the discussion section about the transport-related errors.

Line 300 replaced: In this study we use the first derivative of the concentration time series as a proxy for the large-scale, spatially-integrated flux. However, this should not be directly interpreted as a measure of the underlying flux fields. The atmospheric transport plays an important role in explaining a significant portion of observed $CO_2$ variations at various surface stations (e.g. Krol et al., 2018; Fu et al., 2015) that will affect any interpretation of the CUP metrics. An extensive study was carried out by Lintner et al. (2006), confirming the importance of atmospheric transport to account for some of the inter-annual variations in $CO_2$ observed at Mauna Loa. Murayama et al. (2007) showed how year-to-year changes in the atmospheric transport create significant inter-annual variations in the downward-zero crossing day (DZCD) of the $CO_2$ seasonal cycle that cannot be neglected. Hence, we recommend that while using the EFD method, the contribution of atmospheric transport at the studied background sites should be evaluated before interpreting and relating the CUP metric to sources/sinks.

R1.5

Technical corrections:
"Curve-fitting" is irregularly hyphenated in the text. It needs to be hyphenated when used as an adjective, for example in Line 6, 16, 19, and so on.
Line 256: "using two different curve-fitting methods " => "using the two different curve- fitting methods" is better.

Thank you for pointing it out, this has been corrected.

Other changes:
- We found a bug in our code. In the residual bootstrapping (Fig 3 manuscript), the resampled residuals were added to the observation, rather than the first fitted observation values. This has been corrected. However, there are no significant changes in the results. The revised manuscript has the corrected values and figures.
- We now use acronyms ZCD replacing "zero-crossing dates" to be consistent with previous studies.
- Figures 8 and 9 and Figures 11 and 12 are grouped.
- Figure 13 has been simplified by not coloring the points by year.

References:

Lintner, B. R., Buermann, W., Koven, C. D., and Fung, I. Y. (2006). Seasonal circulation and mauna loa $CO_2$ variability. Journal of Geophysical Research: Atmospheres, 111(D13).

Murayama, S., Higuchi, K., and Taguchi, S. (2007). Influence of atmospheric transport on the inter-annual variation of the $CO_2$ seasonal cycle downward zero-crossing. Geophysical Research Letters, 34(4). ¶

Barlow, J. M., Palmer, P. I., Bruhwiler, L. M., and Tans, P.: Analysis of CO2 mole fraction data: first evidence of large-scale changes in CO2 uptake at high northern latitudes, Atmospheric Chemistry and Physics, 15, 13 739–13 758, https://doi.org/10.5194/acp-15-13739-2015, 2015.¶

Fu, Q., Lin, P., Solomon, S., and Hartmann, D. L.: Observational evidence of strengthening of the Brewer-Dobson circulation since 1980, Journal of Geophysical Research: Atmospheres, 120, 10,214–10,228, https://doi.org/https://doi.org/10.1002/2015JD023657, 2015. gml.noaa.gov: Trends in CO2, [online] Available from:https://gml.noaa.gov/ccgg/trends/, accessed: 2022-06-8.

Krol, M., de Bruine, M., Killaars, L., Ouwersloot, H., Pozzer, A., Yin, Y., Chevallier, F., Bousquet, P., Patra, P., Belikov, D., Maksyutov, S.,Dhomse, S., Feng, W., and Chipperfield, M. P.: Age of air as a diagnostic for transport timescales in global models, Geoscientific Model Development, 11, 3109–3130, https://doi.org/10.5194/gmd-11-3109-2018, 2018.

Wang, X., Xiao, J., Li, X., Cheng, G., Ma, M., Zhu, G., Altaf Arain, M., Andrew Black, T., and Jassal, R. S.: No trends in spring and autumn phenology during the global warming hiatus, Nature Communications, 10, 2389, https://doi.org/10.1038/s41467-019-10235-8, 2019.

---

## Author Comment (AC2)

Response to the reviews of manuscript amt-2022-179: "Reducing errors on estimates of the carbon uptake period based on time series of atmospheric $CO_2$ " by Theertha Kariyathan, Wouter Peters, Julia Marshall, Ana Bastos, Pieter Tans, and Markus Reichstein, to Atmos. Meas. Tech.

Questions from the reviewers are written in blue, our answers in black, text copied from the manuscript is written in *italic*, and all changes in the manuscript are typed in red. When referencing page and line numbers, we are always referring to the old version of the manuscript.

During the review process we came across a study by Barlow et al., 2015, where the CUP is estimated using the first derivative approach. Although we developed our approach independently, we no longer claim the novelty of this approach. We rather emphasize the ensemble approach for uncertainty estimation and rename our method to EFD (ensemble of first-derivative method) and include an extensive discussion of Barlow et al., 2015 in our manuscript. The main changes are made to the introduction (line 65) and discussion (line 270).

Answers to Reviewer 2

This study presents a curve-fitting method and an ensemble-based approach for quantifying the carbon uptake period (CUP; onset, termination and duration) from atmospheric $CO_2$ measurements. The authors have applied the technique to a handful of sites in the Northern hemisphere and shown that the uncertainty associated with the onset and termination of CUP is less with their proposed approach relative to more traditional techniques prevalent within the community. While the illustrations are high-quality, the scientific relevance and the overall flow of the manuscript needs to be improved. Right now, the manuscript reads like a collection of results based on investigations that were conducted and a figure and text to support the investigation. It does not dig deep into the implication of some of the findings (for e.g., Figure 13 is fascinating from a carbon cycle perspective but not explained in any great detail). In addition, the authors have applied their approach to only one seasonal cycle metric and it is not clear if the proposed alternative can be applied to other metrics. There are also inherent assumptions related to the first derivative method that require additional investigations. Along with my comments below, I have suggested a few basic analyses and additional sensitivity test that will improve this study and make it scientifically robust and appealing to the larger carbon cycle science community. I sincerely hope that the authors consider these suggestions. for improving the manuscript.

We thank the reviewer for the critical comments and questions raised. We believe that by answering these questions, the interpretation and portrayal of our results has been improved. Overall, the results and discussion section were restructured and partly re-written to explain our results more clearly. We focus now on the utility of the ensemble-based approach to quantify the uncertainty in the estimation of the CUP using the first-derivative method.

R2.1
Line 1 in the Abstract should read – 'High-quality, long time series measurements of ...'
Thanks, Line 1 is modified as suggested.
***Abstract.*** *High-quality, long time series measurements of atmospheric greenhouse gases show interannual variability in the measured seasonal cycles.*

R2.2 (This comment is broken down to 3 parts, which are addressed separately.)

R2.2.1

Lines 9 – 10: It is a bit misleading to claim that that the approach has been applied to analyze different seasonal cycle metrics as well as claims about the novelty of the approach. The authors have implemented this approach for quantifying one seasonal cycle metric, i.e., the carbon uptake period and associated parameters. What other metrics can be robustly calculated using this approach? It would be extremely relevant to include this in the discussion section.

We agree with the reviewer that the formulation was inaccurate. Here, the ensemble-based approach has been applied to only the CUP and associated parameters, however the approach can also be used to quantify uncertainty in other seasonal cycle metrics for example the seasonal cycle amplitude. Since this is not demonstrated in the study, line 9-10 has been modified:

Line 9-10: We use this ensemble-based approach to analyze the carbon uptake period (CUP: the time of the year when the $CO_2$ uptake is greater than the $CO_2$ release): its onset, termination and duration.

Moreover, we added a sentence on how the method can be applied to other metrics in the discussion (added in Line 320):

Line 320: *In this study we show that $CO_2$ seasonal cycle metric estimates can be strongly sensitive to the method used, hence any method must be thoroughly evaluated before it can be used to draw conclusions from the data.* In Barlow et al. (2015) the robustness of the first derivative is tested by evaluating its ability to capture a known trend from a synthetic time-series. The synthetic time-series were given a linear trend and interannual variations in peak uptake of ±10 days, allowing their method to retrieve the ensemble members provide an uncertainty range, hence allowing the robustness of estimated values to be estimated. the true linear trend to within 10-25%. Our EFD-approach provides uncertainty on the year to year variability in the seasonal cycle metrics based on the fitted data residuals, which could be used in a trend analysis to give differential weights to each year. Also, trend analysis on the individual ensemble members would allow uncertainty on the trend to be calculated. Our demonstration of the EFD-method on the CUP could be extended to other metrics that are derived directly from the seasonal cycle in a similar way, for example the peak to trough amplitude especially when curve-fitting discrete data, or at sites with broad or multiple peaks. In a similar fashion, the ensemble-based approach could be used to evaluate a newly proposed method or select an optimal method for evaluating any other metric based on reduced uncertainty.

R2.2.2
Right now, the Discussion section reads more like a collection of results than a true Discussion that provides scientific implications (see also comment #7) and relevance of this method for the carbon cycle community.
This will be addressed along with comment 7 (R2.7)

R2.2.3
In addition, the technique proposed by the authors are not new per se, but its application for quantifying the seasonal cycle metric is novel – the authors need to clearly distinguish this throughout the manuscript.
Thank you for pointing it out. The parts of texts which claim the novelty of the methods used here have been modified (e.g., Lines 7, 64, and 264). Further changes were made to address a similar comment from Reviewer #1 (See R1.2). We also include an extensive discussion of Barlow et al., 2015, a study that had previously introduced the first-derivative approach.

R2.3
Lines 21 – 22 – the statement is applicable to not just measurements made at Mauna Loa, but almost all atmospheric $CO_2$ measurements, be it in situ or remotely-sensed. Kindly rephrase to either make it more generic or more specific to Mauna Loa.
Thank you, we agree that this should be corrected. Lines 21–22 have been modified based on this comment as follows:

Line 21: Ongoing in-situ measurements of the atmospheric $CO_2$ mixing ratio have revealed an increase in $CO_2$ mole fraction in the atmosphere. *The increase in atmospheric $CO_2$ due to release of carbon from fossil fuel burning and land-use change is buffered by net $CO_2$ uptake by the ocean and land biosphere (Keeling, 1960).*

R2.4
Line 50 – The authors should be more specific about which metrics they are talking about and specify the ones that are highly sensitive to data gaps or noise in the time- series.

Thank you for the comment. An example is given in Line 51 (*One example is the timing of the carbon uptake period (CUP)*), but the lines were rephrased to improve clarity. Lines 50 -55 have been modified as follows:

Metrics derived from $CO_2$ time series such as the seasonal cycle peaks can be highly sensitive to data gaps and noise. This is especially true for metrics associated with the growing season onset at higher latitude sites, where $CO_2$ time series show flat or multiple peaks in winter (Barlow et al., 2015). Hence, deriving other metrics like the timing of the carbon uptake period (CUP) from the seasonal cycle maximum results in less robust estimates.

R2.5
Line 69 – 70 - What about checking an alternate approach? In the Introduction, the authors made the argument that multiple approaches should be tested. Why haven't they implemented that rationale here?
We agree that the lines are confusing. Here, we want to test the robustness of the EFD method. We want to understand if the low uncertainty while using the EFD method is dependent on the specific curve-fitting method used here. This is why we use another curve-fitting method and test the EFD method using both. We observe that, for both curve-fitting methods used, the EFD method leads to lower spread in the estimated CUP.

We corrected the sentence as follows:

Line 69 : We also tested if the EFD method is sensitive to the specific curve-fitting method applied by fitting the data with the commonly-used CCGCRV method, which is a frequency-domain-based filter, similar to the wavelet transform approach of Barlow et al. (2015).

R2.6

Lines 146-152, Page 7 – A big assumption in implementing the FDT approach is that "the first derivative of the $CO_2$ dry air mole fraction is a proxy for the flux", thereby completely ignoring the role of atmospheric transport. This is especially relevant as the majority of sites the authors have selected are the marine boundary layer sites, which are designed to sample the background flux and not necessarily changes in local flux. Can the authors demonstrate the robustness of their assumption by doing pseudo-data / simulated data experiments? For example, the authors can use known fluxes from CarbonTracker or CarbonTracker-Europe, generate pseudo-data at the sites used in the study, and demonstrate that the first derivative is indeed an approximation of the flux signal.

Thank you, this is a very important point. We consider the first derivative to be a proxy for the seasonal cycle of hemispheric-scale NEE, not a one-to-one measure of local fluxes. The seasonal variability in atmospheric $CO_2$ at background sites should reflect the spatial integral of the fluxes over large latitudinal or hemispheric scales, but the area of integration is affected by atmospheric transport, especially at marine

boundary sites as mentioned by the reviewer. To address the reviewer's concern, we performed a synthetic data experiment using known NEE from the Jena CarboScope inversion to test the accuracy of the EFD method in deriving a prescribed change in CUP.

We did simulations in which idealized NEE fluxes were transported forward and the atmospheric concentrations were sampled at the location of the measurement sites. In the baseline simulation, a fixed year from the Jena CarboScope Inversion (Rödenbeck et al., 2003) (version ID: sEXT ocNEET v2021) was used to generate an idealized NEE flux time_-series that has no inter-annual variability (IAV) in the CUP_NEE (CUP of the NEE flux) at any given pixel. Then, we prescribe changes to the CUP_NEE at Northern Hemisphere land pixels with clear seasonal cycles by steps of -10,-8...+8,+10 days, creating different ΔCUP_NEE scenarios (change from baseline CUP_NEE). The fluxes were transported forward using the atmospheric transport model TM3 (Heimann and Körner, 2003) with wind fields from a fixed year (to remove transport IAV) and the resulting ΔCUP_MR (change of CUP_MR (CUP of the simulated $CO_2$ mixing ratio) from the baseline simulation) was compared to the ΔCUP_NEE. We calculate ΔCUP_MR using both the EFD method and zero-crossing method and compare their sensitivity to ΔCUP_NEE.

[Figure]

*Figure 1: The boxplot shows the ΔCUP_MR over the studied years estimated using the zero-crossing method (blue) and the EFD-method (orange) at MLO. 'α' denotes the slope of the regression line fitted to the median of the boxplot.*

We find that the EFD-estimated ΔCUP_MR has a strong linear relationship to the applied ΔCUP_NEE, but it returns ΔCUP_MR by a factor smaller than what was applied in flux space. This factor depends on the station, as for MLO (shown in the figure), it's quite close to a factor of 2. The zero-crossing-estimated ΔCUP_MR has weaker one-to-one relation relative to EFD estimation which shows that the zero-crossing method is relatively less sensitive to the changes in the "actual" CUP. These differences are shown for the station MLO in Figure 1. This implies that the EFD-derived CUP is likely a more robust metric than the

zero-crossing--approximated CUP, but indeed should not be interpreted as a direct one-to-one signal of the underlying flux field. Transport to the sites, and mixing of spatially varying NEE signals with differing CUP timing, integrate to a reduced atmospheric expression of CUP changes in biospheric fluxes. That integration depends on the site location, which is what we will consider next. We believe this requires a more detailed study about the influence of transport on signals received at the background sites, which is not within the scope of this study. Therefore, we have modified the discussion to indicate that the role of transport should be considered when studying the observed $CO_2$ seasonal cycle.

Line 300 replaced: In this study we use the first derivative of the concentration time series as a proxy for the large-scale, spatially-integrated flux. However, this should not be directly interpreted as a measure of the underlying flux fields. The atmospheric transport plays an important role in explaining a significant portion of observed $CO_2$ variations at various surface stations (e.g. Krol et al., 2018; Fu et al., 2015) that will affect any interpretation of the CUP metrics. An extensive study was carried out by Lintner et al. (2006), confirming the importance of atmospheric transport to account for some of the inter-annual variations in $CO_2$ observed at Mauna Loa. Murayama et al. (2007) showed how year-to-year changes in the atmospheric transport create significant inter-annual variations in the downward-zero crossing day (DZCD) of the $CO_2$ seasonal cycle that cannot be neglected. Hence, we recommend that while using the EFD method, the contribution of atmospheric transport at the studied background sites should be evaluated before interpreting and relating the CUP metrics to sources/sinks.

R2.7 and R2.2.2
Section 5 – Discussion – other than a few segments, this section seems to be a continuation of the previous section. The authors need to rethink the way they present this section, move the results to the previous section and/or focus more on the scientific implications of their findings. It would also be useful to dig deep into a couple of the results and talk about the scientific findings rather than present one result after the another.

Thank you for the suggestion. The separation of the results and discussion section has been improved. To do this we have reordered some of the sections, separating the results from the discussion accordingly. A summary of the resulting structure is given below.
Results
- Comparison of the three CUP calculation metrics is presented in Figures 6, 7, 8 and 9.
- Fig 13 (now Fig. 8) and its explanation (Line 297) and additional observation that: The X-axis range, showing the CUP from ZCD in Fig.8, is unlikely to represent the "actual" year-to-year variation in the CUP, with the largest variation seen at MLO, NWR and MID. (This will be further explained in the discussion)
- Further testing using CCGCRV fitting, presented in Figure 10 and 11.

Discussion

In the Discussion we draw upon Figures 5 and 10 to interpret the figures presented in the Results section above. We have further included the other modifications following the reviewer comments. Figure 10 shows that years with a similar duration of the "actual" CUP can have different CUP duration when determined with the zero-crossing method, explaining the large year--to--year variation in the X-axis of Fig. 13.

R2.8
Figure 13 - What are the conclusions from this figure? Do we show any important trends? Any relevance to carbon cycle science? Similar to the previous comment, this seems another missed opportunity to delve

deeper into the results and provide scientific implications and context for the results. I would strongly recommend the authors to select a few key results and figures, and then delve deeper into them rather than presenting all results and figures generated during their investigation.

Thank you for these questions. Analyzing Fig 13 in light of these questions improves our understanding of the obtained results. Our observations are summarized as follows and will be included in the discussions.

We find that in addition to having a larger annual uncertainty, the ZCD-approximated CUPs have a larger range over the study period compared to the EFD-estimated CUPs (Fig 8). For example, at MLO the zero-crossing--approximated CUP ranges from 100 to 250 days, corresponding to a difference of 3-8 months. This is unrealistically large, considering that (a) MLO receives signals mainly from North America and Eurasia (Buermann et al., 2007), where the growing season has lengthened on average by 2.6 days per decade (Park et al., 2016), and (b) phenology statistics indicate that at MLO the average CUP of 155 days varies by ±17.4 days between 1969 to 2013 (Gonsamo et al., 2017), a variation that is 10x smaller than the ZCD-derived one. The ZCD includes changes in the both latter part of the net uptake period and the early release period, making it difficult to separate the contribution of the net uptake and net release periods to the changes in the CUP estimate. To understand this large spread in CUP, we compare two years with different ZCD approximated CUP at MLO, shown in Fig 11. We find that the difference in the CUP estimate in this case is due to the change in the early release period, whereas the uptake periods are essentially the same. The EFD method, by definition, is not affected by differences in the net release period and can therefore provide a more robust metric of CUP duration.

Atmospheric transport can contribute to the IAV in CUP estimates while using both the EFD and ZCD. However, the ZCD is influenced by transport variability in both the late uptake and early release periods. Hence, changes in the early release period could be erroneously interpreted as changes in the CUP when using the ZCD. Years with extreme CUP approximated by the ZCD suggest that there is reduced net respiration in the early release period, thereby prolonging the time to reach the UZCD. This is determined by the interplay of the $CO_2$ uptake and release processes, which are influenced by physical factors like temperature, soil moisture and solar radiation. For example, in dry conditions there is less respiration by plants and slower decomposition of organic matter in the soil, resulting in reduced $CO_2$ release to the atmosphere (Yan et al., 2018). The rate of decomposition further depends on the snow cover and available detritus content in the soil following leaf senescence. Furthermore, in the early release period, when the solar radiation is not limited, plants may continue to photosynthesize depending on water availability and temperature, leading to reduced net $CO_2$ release. Thus, in years with extreme CUP as approximated by the ZCD, the physical processes that affects the release period should be investigated. In comparison to the CUP definition, the approximation by the ZCD is also sensitive to variations after the summer minimum, i.e. during the early release period. A more thorough investigation of the sensitivity of the EFD and ZCD to CUP interannual variability would require dedicated modelling experiments, which is beyond the scope of the current study.

Other changes:
- We found a bug in our code. In the residual bootstrapping (Fig 3 manuscript) the resampled residuals were added to the observations, we corrected this by adding the residuals to the first fitted observation values. However, there are no significant changes in the results. The revised manuscript has the corrected values and figures.
- We now use acronyms ZCD replacing "zero-crossing dates" to be consistent with previous studies.
- Figure 8, 9 and Figures 11,12 are grouped.
- Figure 13 is simplified by not coloring the points.

References:

Lintner, B. R., Buermann, W., Koven, C. D., and Fung, I. Y. (2006). Seasonal circulation and mauna loa $CO_2$ variability. Journal of Geophysical Research: Atmospheres, 111(D13).

Murayama, S., Higuchi, K., and Taguchi, S. (2007). Influence of atmospheric transport on the inter-annual variation of the $CO_2$ seasonal cycle downward zero-crossing. Geophysical Research Letters, 34(4).

Barlow, J. M., Palmer, P. I., Bruhwiler, L. M., and Tans, P.: Analysis of CO2 mole fraction data: first evidence of large-scale changes in CO2 uptake at high northern latitudes, Atmospheric Chemistry and Physics, 15, 13 739–13 758, https://doi.org/10.5194/acp-15-13739-2015, 2015.

Fu, Q., Lin, P., Solomon, S., and Hartmann, D. L.: Observational evidence of strengthening of the Brewer-Dobson circulation since 1980, Journal of Geophysical Research: Atmospheres, 120, 10,214–10,228, https://doi.org/https://doi.org/10.1002/2015JD023657, 2015. gml.noaa.gov: Trends in CO2, [online] Available from:https://gml.noaa.gov/ccgg/trends/, accessed: 2022-06-8.

Krol, M., de Bruine, M., Killaars, L., Ouwersloot, H., Pozzer, A., Yin, Y., Chevallier, F., Bousquet, P., Patra, P., Belikov, D., Maksyutov, S.,Dhomse, S., Feng, W., and Chipperfield, M. P.: Age of air as a diagnostic for transport timescales in global models, Geoscientific Model Development, 11, 3109–3130, https://doi.org/10.5194/gmd-11-3109-2018, 2018.

Gonsamo, A., D'Odorico, P., Chen, J. M., Wu, C., and Buchmann, N.: Changes in vegetation phenology are not reflected in atmospheric CO2 and 13C/12C seasonality, Global Change Biology, 23, 4029–4044, https://doi.org/https://doi.org/10.1111/gcb.13646, 2017.

Buermann, W., Lintner, B. R., Koven, C. D., Angert, A., Pinzon, J. E., Tucker, C. J., and Fung, I. Y.: The changing carbon cycle at Mauna Loa Observatory, Proceedings of the National Academy of Sciences, 104, 4249–4254, https://doi.org/10.1073/pnas.0611224104, 2007.

Park, T., Ganguly, S., Tømmervik, H., Euskirchen, E. S., Høgda, K.-A., Karlsen, S. R., Brovkin, V., Nemani, R. R., and Myneni, R. B.: Changes in growing season duration and productivity of northern vegetation inferred from long-term remote sensing data, Environmental Research Letters, 11, 084 001, https://doi.org/10.1088/1748-9326/11/8/084001, 2016.

---

## Author Response (AR2)

Comments from the reviewer and editor are in red text and the response is in blue text.

Editor's comments:
I would strongly recommend that the authors consider some of the recommendations provided by the reviewer. From the revised submission, it becomes clear that the FDT method has been applied before. However, this does not lessen the scientific value and significance of this work, which makes it highly relevant the carbon cycle science community. The manuscript would be made much stronger and valuable if the authors at a minimum provide a more detailed quantitative comparison with a figure or table to demonstrate that the FDT approach is improved by adding the ensemble part. This goes beyond the fact that a clear advantage of the EFD approach is the ability to calculate uncertainties as shown in Section 5.1. And/or expand more on the advantages of their method over the ZCD method. All of this can be handled via a short discussion section - again pieces of which are scattered in the text and Conclusions. Overall, the authors have done a commendable job in handing the reviewer comments and suggestions from the original version and I look forward to seeing the impact the proposed methods have on improving our understanding of the carbon uptake period.

Response:
Thank you for suggesting the resubmission of the manuscript with revisions. The suggestion was to strengthen the comparison between the EFD and the approach in Barlow et al., (2015) (hereafter referred to as Barlow) or between the EFD and the ZCD method. We address this in a new section (Section 5.3) where we test the sensitivity of the three methods (ZCD, EFD and the approach in Barlow) to actual changes in the CUP. We compare the three methods based on their ability to identify changes made in the CUP of idealized NEE fluxes. The ZCD method is found to be the least sensitive, adding to the advantage of methods based on the first derivative of the $CO_2$ time series over the ZCD method, in addition to those discussed previously. We do not find substantial differences between the sensitivity of EFD and the approach in Barlow. However, the approach in Barlow does not give an uncertainty estimate. Hence the uncertainty in its sensitivity could not be evaluate, clearly showing a disadvantage of the method. Thus, the sensitivity analysis strengthens our claim that the EFD method adds to the benefits of the method in Barlow and gives robust estimates than the ZCD method.

Reviewer's comment:
In my initial review of the paper, I found the first derivative threshold (FDT) method for estimating the carbon uptake period (CUP) to be the interesting aspect of this study. It is unfortunate that the authors were not aware that this method had been previously published by Barlow et al. (2015). While the authors have acknowledged this multiple times in the revised manuscript and provided some discussion, the lack of the novelty of the FDT method weakens the paper's suitability for publication in AMT. To make the paper stronger, the authors need to provide some additional analysis.

The paper needs to provide a clearer explanation of the advantages of using an ensemble threshold approach with FDT. There is very limited discussion and no figures or tables comparing their ensemble first derivative (EFD) and FDT methods. Instead, the paper extensively compares the performance of the EFD method and the zero-crossing dates (ZCD) method. The authors hint that the ensemble approach is better than the FDT method, but these claims need to be backed by quantitative analysis, including a figure. Additionally, the paper needs to explain in detail how the data-derived year-to-year uncertainty from their ensemble approach provides a more robust threshold estimate, and how

this could improve trend analyses of seasonal cycle changes in the context of understanding the climate cycle.

Given that Barlow (2015) has already shown that the FDT method provides a more robust estimate of the key dates that define the CUP than the ZCD method, the authors need to clearly state what additional advantages of the EFD method over the ZCD they have identified in their analysis.

The most interesting analysis in the study is in response to reviewer 2's comment. The authors show, using a transport model run, that the mixing ratio CUP derived from the ZCD method is much less sensitive to flux CUP than the mixing ratio CUP from the EFD method. This analysis is very insightful, although it would have been better if the analysis had been conducted at multiple measurement sites instead of just MLO and if the inter-annual variability in the transport model had not been removed.

In conclusion, I recommend major revisions of the paper with a focus on one or more of the following points:

Thank you for the detailed review. We agree that the manuscript would be improved through the addition of a more quantitative comparison between the different methods. We carried out a synthetic data experiment which addresses these points, as described below.

1. Provide a more detailed quantitative comparison with a figure or table to demonstrate that the FDT approach is improved by adding the ensemble part.

In response, we added a new figure (Fig 12) and section (5.3) comparing the sensitivity of the different methods to changes in the actual CUP. We performed a synthetic data experiment where known changes in the CUP were imposed on idealized NEE fluxes and we simulated $CO_2$ time series using an atmospheric transport model. The different CUP estimation methods were evaluated for their ability to capture the imposed CUP changes from the simulated time series, as in the previous response to Reviewer 2. We find that the EFD method and the approach in Barlow have similar sensitivities to imposed CUP changes. However, the uncertainty in sensitivity for the approach in Barlow, could not be evaluated as the method does not give an uncertainty range for its estimates, showing the advantage of the ensemble approach to constrain uncertainties in metrics.

2. Use transport model analysis to demonstrate how the application of the FDT/EFD method on mixing ratios is better than the ZCD method to determine CUP of fluxes.

From the experiment described above, we find that the CUP estimation methods based on the first derivative of the $CO_2$ time series (i.e., the EFD and approach in Barlow) are more sensitive to CUP changes than the ZCD method. This can be seen from the mean slope of the fitted regression lines in Fig. 12.

3. Clearly state the new advantages of the FDT/EFD method over the ZCD method found in this study, in addition to those stated by Barlow (2015).

The sensitivity experiment demonstrates the advantage of the EFD method over the ZCD method more conclusively than what was presented by Barlow. Not only is the ZCD method less sensitive to changes in the CUP, it also has higher uncertainties.

4. Show how a tighter constraint on the mixing ratio CUP using the EFD method leads to a reduction in errors in flux CUP.

As seen by the slopes of the lines in Fig. 12, none of the methods are able to estimate the true length of the CUP or the changes therein. The 15% threshold was chosen in order to minimize the uncertainty with the onset time (as seen in Fig. 5) while with minimal truncation of the CUP. Figure 12 compares only the changes in the CUP inferred from the mixing ratio (i.e., $\Delta$CUP_MR): when comparing the absolute length, we find that CUP_MR is, on average, 5 days longer when using the 15% threshold compared to the 25% threshold.

Other comments:

Please complete your list of affiliations by including country names.
Thank you, the country names are included in the list of affiliations.

Please note that, there is a change in the order of the list of authors.
Previous: Theertha Kariyathan[1,2], Wouter Peters[2], Julia Marshall[3], Ana Bastos[1], Pieter Tans[4], and Markus Reichstein[1]
Current: Theertha Kariyathan[1,2], Ana Bastos[1], Julia Marshall[3], Wouter Peters[2], Pieter Tans[4], and Markus Reichstein[1]

Reference:

Barlow, J. M., Palmer, P. I., Bruhwiler, L. M., and Tans, P.: Analysis of $CO_2$ mole fraction data: first evidence of large-scale changes in $CO_2$ uptake at high northern latitudes, Atmospheric Chemistry and Physics, 15, 13 739–13 758, https://doi.org/10.5194/acp-15-13739-2015, 2015.